# Hormonal Determinants of Growth and Weight Gain in the Human Fetus and Preterm Infant

Laura Page [1], Noelle Younge [2] and Michael Freemark [1,3,*]

1 Division of Pediatric Endocrinology, Duke University Medical Center, Durham, NC 27710, USA; laura.page@duke.edu
2 Neonatology, Duke University Medical Center, Durham, NC 27710, USA; noelle.younge@duke.edu
3 The Duke Molecular Physiology Institute, Duke University Medical Center, Durham, NC 27710, USA
* Correspondence: michael.freemark@duke.edu

**Abstract:** The factors controlling linear growth and weight gain in the human fetus and newborn infant are poorly understood. We review here the changes in linear growth, weight gain, lean body mass, and fat mass during mid- and late gestation and the early postnatal period in the context of changes in the secretion and action of maternal, placental, fetal, and neonatal hormones, growth factors, and adipocytokines. We assess the effects of hormonal determinants on placental nutrient delivery and the impact of preterm delivery on hormone expression and postnatal growth and metabolic function. We then discuss the effects of various maternal disorders and nutritional and pharmacologic interventions on fetal and perinatal hormone and growth factor production, growth, and fat deposition and consider important unresolved questions in the field.

**Keywords:** insulin; insulin-like growth factors; growth hormone; lactogenic hormones; glucocorticoids; adipocytokines

## 1. Introduction: The Metabolic Risks of Preterm Birth

Preterm birth and low birth weight are the major causes of morbidity and mortality during the first year of life and predispose to adiposity, hypertension, insulin resistance, glucose intolerance, and type 2 diabetes in adulthood. A recent meta-analysis [1] of 43 studies comparing 18,295 adults born preterm with 294,063 adults born at term found that prematurity was associated with future increases in percent body fat mass (+1.2–1.5%), systolic and diastolic blood pressures (4.7 and 2.3 mm Hg, respectively), and fasting insulin (+16%) and glucose (+0.06–0.07 mmol) levels. Rates of type 2 diabetes correlate inversely with gestational age and birth weight, with 30–150% higher risks in adult men and women who had birth weights less than 2 kg [2–5]. The risks of long-term metabolic complications are highest in former preterm, small-for-gestational-age infants who have excessive postnatal weight gain.

Rates of prematurity are sex-dependent, more common in pregnancies with a male fetus. Interestingly, there appear to be sex differences in the rates of postnatal growth of former preterm infants and in the long-term metabolic complications of preterm birth: for example, weight z-scores at 20 years of age are greater in girls than in boys born preterm [6], and the relative risk of adult type 2 diabetes in females born with low birth weight exceeds that in former low birthweight males [4,7]. Disparities in risk may be related in part to effects of sex steroids in utero and/or sex differences in fetal and postnatal growth and weight gain, muscle mass, and fat deposition and distribution [8].

The short- and long-term metabolic complications of prematurity result from incomplete or impaired organ development, dysregulation of metabolic functions, and adaptations in hormonal pathways that promote adiposity and limit the production and action of insulin. Contributing factors in the preterm infant include:

(a) **Inadequate reserve of skeletal myocytes and pancreatic beta cells**, which undergo neogenesis and proliferation primarily in late gestation and early infancy [9–11]. A relative reduction in skeletal myocytes in preterm infants reduces muscle insulin sensitivity [12–14], limits future energy expenditure [15,16], and contributes to relative adiposity [17], while a reduction in beta cell mass, delayed maturation of glucose transporter 2, and defective processing of proinsulin [10,11,18–21] attenuate (in those born AGA, not SGA) the insulin secretory response to insulin resistance.

(b) **Incomplete maturation of the vascular tree** [22] and **a deficiency of renal nephrons** [23–25], a majority of which are formed in late gestation and the early postnatal period. In combination, these may impair endothelial and renal function [26] and predispose to future hypertension [17,27–30].

(c) **Deficiencies of hormones and growth factors** that normally emerge or increase in late gestation (see discussion below).

(d) **Adaptive hormonal responses** to preterm delivery and various intrauterine and perinatal insults including hypoxia, malnutrition, cardiorespiratory and gastrointestinal disorders, neurologic insults, and infectious diseases (see discussion below).

(e) **Effects of various therapeutic agents including glucocorticoids**, which impede skeletal muscle and bone growth and reduce tissue insulin sensitivity (see discussion below).

The absolute risk of future metabolic dysfunction in a premature child depends critically on additional factors including genetic inheritance and the family history of obesity, hypertension, and type 2 diabetes, the rates of catch-up weight gain and linear growth during the perinatal and prepubertal periods [31–33], and the relative changes in lean and body fat mass and fat distribution during adolescence and adult life [34–37].

## 2. Methodologic Approach to the Narrative Review

The long-term metabolic complications of prematurity derive in part from an imbalance of fat deposition and linear growth that begins during early postnatal life. As discussed in detail in the narrative below, this results from developmental immaturity of fetal and neonatal hormone production and action and a hormonal adaptive response that promotes survival during acute illness. Thus, an understanding of the pathogenesis of metabolic dysfunction necessitates an in-depth analysis of the hormonal determinants of growth and weight gain in the human fetus and preterm infant.

To that end, we undertook a comprehensive review of the literature (in English) published between January 1967 and August 2023. Publications were identified using a systematic search of the PubMed database. Given the broad scope of the discussion and the notable differences in fetal growth and fat storage among species, our review largely focused on human studies. Nevertheless, groundbreaking investigations in experimental animals were also analyzed. All publications were evaluated critically for scientific rigor, impact, and relevance to our subject.

In this narrative review, we assess the dynamic changes in human linear growth, weight gain, lean body mass, and fat mass during mid- and late gestation and the early postnatal period in light of changes in maternal, placental, and fetal hormones, growth factors, and adipocytokines. We characterize the effects of hormonal determinants on placental nutrient delivery and fetal nutrient utilization and the consequences of preterm delivery for hormone expression and newborn growth and metabolic function. We discuss the effects of various maternal disorders and pharmacologic interventions on fetal and perinatal hormone and growth factor production, growth, and fat deposition. Finally, we consider major gaps in knowledge and identify potential new targets for future assessment and treatment.

Except when particularly relevant (for example, development/function of the placenta), we discuss general body, not tissue-specific, growth and weight gain. We defer any detailed discussion of specific syndromes or genetic disorders causing abnormalities



in growth and/or weight gain, except to illustrate the physiological roles of hormones, growth factors, and adipocytokines in normal development.

Levels of hormones, growth factors, and cytokines depicted in the various figures have been adapted from representative published studies. It should be noted that new advances in assay technology make the *absolute* values depicted in the figures less reliable than the *trends* and *patterns* of changes in hormone, growth factor, and adipocytokine levels during development. Whenever possible, we have prioritized studies of fetal hormones, growth factors, and cytokines in blood samples obtained by *cordocentesis*; we assume that the stress of delivery and separation of the placenta make levels of certain hormones (for example, placental hormones, cortisol, and prolactin) in *ex utero* cord or neonatal blood less representative and more difficult to interpret than levels obtained in utero by cordocentesis. Nevertheless, the levels of hormones and cytokines in cordocentesis samples are in most cases comparable to those obtained in umbilical cord blood at similar gestational age.

### 3. Rates of Linear Growth, Weight Gain, and Fat Deposition during Fetal and Early Postnatal Development

Standard clinical growth curves depicting changes in length and weight during the course of fetal and postnatal development have important limitations. First, measurement of total length during early development does not provide an accurate metric of linear growth, given that the head constitutes a disproportionately large percentage of overall body size. Second, measurements of accrued length may obscure critical changes in the *rate* of linear growth. Third, measurements of body weight provide no information regarding the *ratio of fat mass to lean body mass*, an important determinant of future metabolic risk. It is therefore useful to assess changes in femur growth and percent body fat as measures of long bone growth and fat deposition, respectively, during early development.

As shown in Figure 1 (left), adapted from data provided by the World Health Organization [38], the rate of femur growth peaks at mid-gestation and declines progressively thereafter. The sharp decrease in growth velocity near term may be related to the late gestational surge in fetal cortisol ([39]; see below).

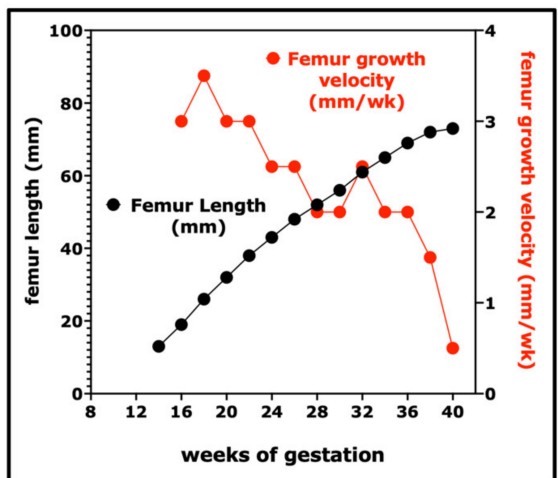 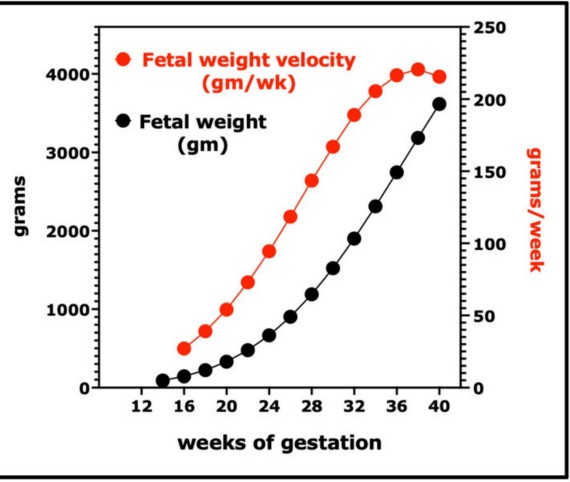

**Figure 1.** Femur growth and weight gain in the human fetus. Figures are adapted from data from the World Health Organization [38]. The data represent the 50th percentile of the combined cohort of boys and girls (the very small differences between the sexes at the 50th percentile are not shown).

In contrast to linear growth, the rate of weight gain (Figure 1, right) increases progressively from mid-gestation to term.

The progressive increases in weight during the second half of gestation and the early postnatal period are associated with a marked accumulation of adipose tissue mass (Figure 2, left) [40,41]. In combination with a decrease in linear growth velocity, this results in a progressive increase in % body fat (Figure 2, right) [40–44].

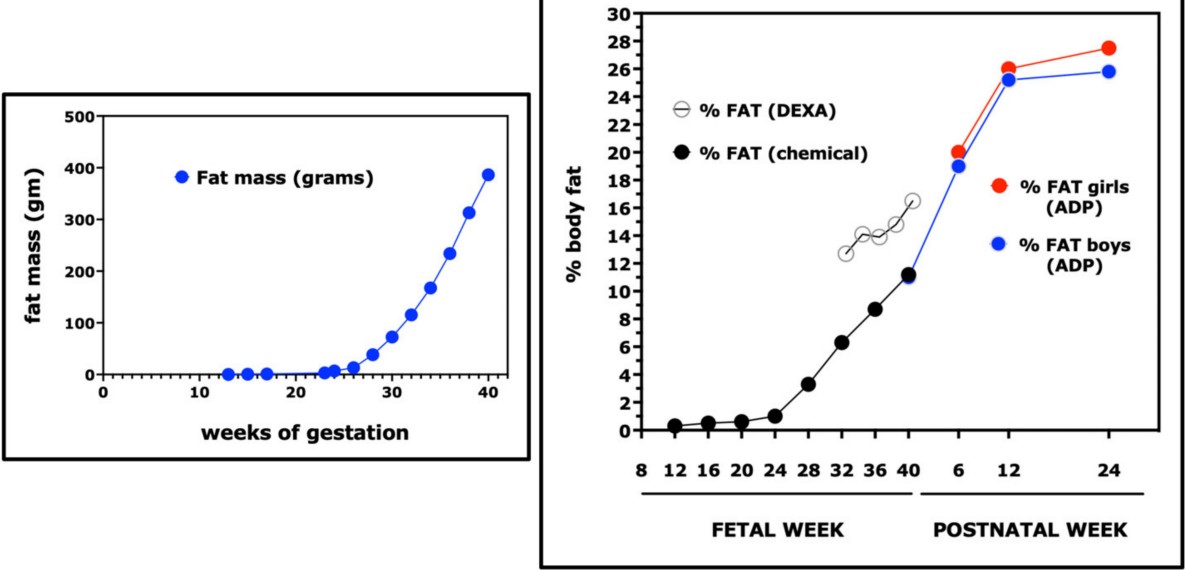

**Figure 2.** Fat deposition in the human fetus and newborn infant. (**left**) The figure is adapted from data in Ziegler EE et al. [40]; qualitatively similar findings in Widdowson EM. [41]. (**right**) The figure is adapted from chemical analysis data from Ziegler EE et al. [40] with qualitatively similar findings in Widdowson EM [41]. DEXA scan estimates of body fat are from Lapillonne A et al. [42]. Representative data obtained using air displacement plethysmography (ADP) in the postnatal period are from Hamatschek C et al. [43] and Murphy-Alford A et al. [44].

Most of the fat accrued by the fetus in late gestation and the early postnatal period has characteristics of white adipose tissue. However, the fetus also stores brown adipose tissue in buccal, interscapular, mediastinal, and perirenal depots. Limited evidence suggests that BAT stores increase over time during the latter half of gestation (Figure 3) and by term may represent 5–7% of total fetal fat content [45,46].

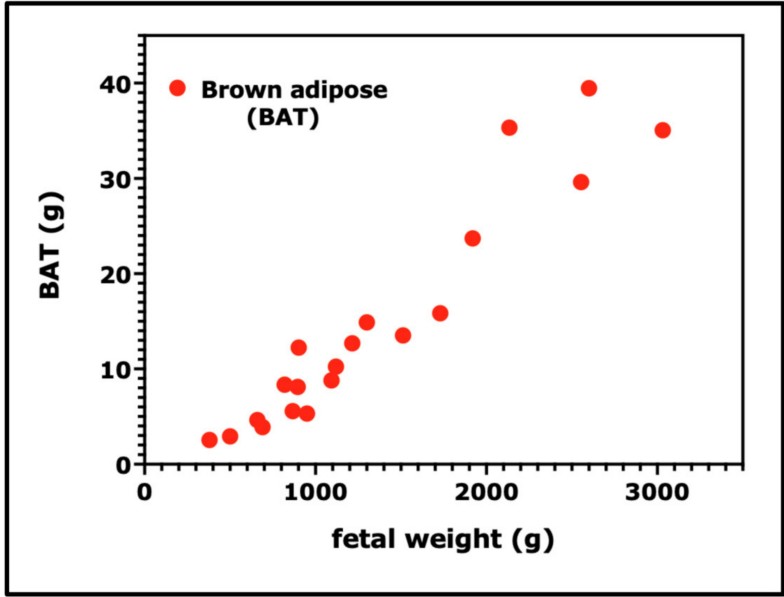

**Figure 3.** Brown adipose stores in the human fetus. The figure is adapted from data in Merklin RJ. [45] and Moragas A and Toran N. [46].

We know very little about the relative contributions of adipogenesis (the generation of new fat cells from mesenchymal pre-adipocytes) and lipogenesis (fatty acid synthesis and

the accumulation of triglyceride within pre-existing fat cells) during human adipose tissue development or the factors regulating the distribution and size of the various fetal and neonatal white and brown adipose tissue depots. It seems likely that fat accrual during mid–late gestation and the early postnatal period is controlled by a complex, stage-dependent interplay of adipogenic hormones (e.g., the insulin-like growth factors, glucocorticoids, and prolactin), lipogenic factors (insulin), and mediators of lipolysis (e.g., growth hormone and the catecholamines). As discussed below, dysregulation of hormone and growth factor production and action in preterm infants can have profound effects on adipose development and linear growth and likely contributes to future metabolic risk.

## 4. Roles of Maternal and Placental Hormones, Growth Factors, and Cytokines in Maternal Metabolism and Transplacental Nutrient Delivery

The substrates of fetal growth and weight gain are nutrients that derive from the pregnant mother, who undergoes striking adaptations to meet fetal demands. Maternal food intake increases early in gestation and nutrients and calcium are stored as energy and mineral reserves for the growing fetus and newborn child. In mid- and late gestation, energy stores are mobilized during fasting to serve the mother's own metabolic demands and to facilitate placental transport of nutrients for fetal development and growth.

As discussed in detail in a previous manuscript [47], these adaptations are orchestrated by striking changes in the production of maternal and placental hormones, cytokines, and growth factors (Figure 4).

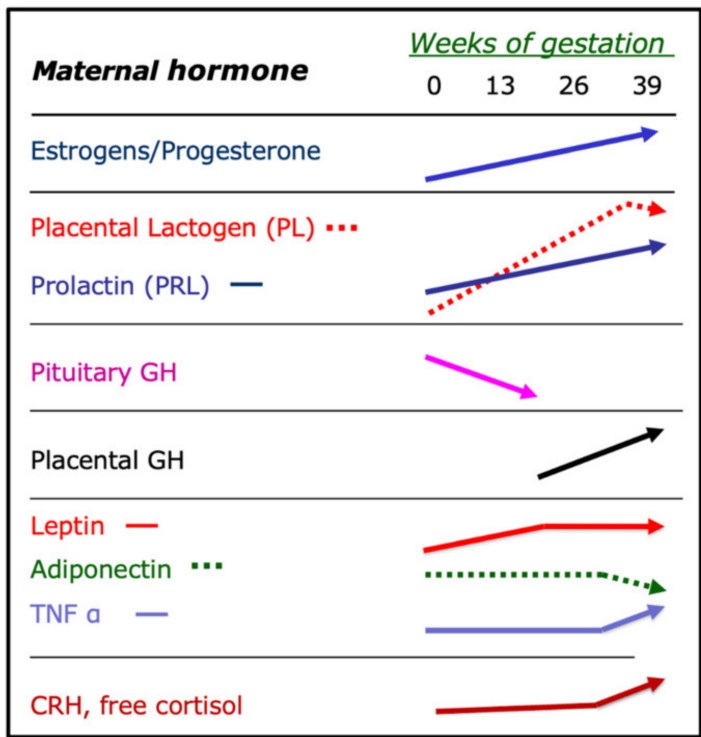

**Figure 4.** Changes in maternal hormones and cytokines during pregnancy. GH, growth hormone; CRH, corticotropin-releasing hormone. Adapted from Newbern D and Freemark M [47].

In early to mid-gestation, there are progressive increases in placental production of sex steroids, human placental lactogen (hPL, also called human chorionic somatomammotropin), and leptin; at the same time, maternal pituitary prolactin levels rise, while pituitary growth hormone (GH) secretion declines. The lactogenic hormones (hPL and prolactin) facilitate food intake and fat deposition during pregnancy and up-regulate intestinal calcium channels TRPV5 and 6 and Calbindin and thereby promote gastrointestinal calcium absorption. The concurrent fall in pituitary GH secretion maintains insulin sensitivity in early gestation despite the rise in progesterone, thereby facilitating maternal fat deposition.

In mid–late gestation, there are progressive increases in sex steroids, hPL, prolactin, TNF alpha, and cortisol and a fall in adiponectin. The levels of pituitary GH (GH-N) in the maternal circulation decline markedly at mid-term, followed by a striking and progressive increase in the levels of placental GH (GH-V). Maternal food intake and fat mass increase further, but maternal metabolism is transformed by the emergence of insulin resistance (IR).

The major determinant of maternal IR in late gestation is the rise in placental GH (Figure 5). In combination with cortisol, inflammatory cytokines, progesterone, and, possibly, hPL [48], placental GH induces lipolysis and insulin resistance in adipose tissue and skeletal muscle, freeing maternal glucose, amino acids, essential fatty acids, and ketones for transport to the fetus. Transplacental nutrient delivery and fetal growth are facilitated by the GH-induced rise in maternal insulin-like growth factor 1 (IGF-1), which, together with estrogen, promotes uterine growth and blood flow. At the same time, leptin and IGF-2, derived from the fetus and the placenta [49,50], and a fall in maternal adiponectin promote placental growth and the transport of glucose, amino acids, and possibly essential fatty acids to the fetus [49,51–53]. In combination, these ensure that the fetus remains adequately nourished during periods of maternal fasting. Indeed, by increasing placental nutrient availability, the secretion of placental GH, and the development of maternal IR, are essential for fetal growth.

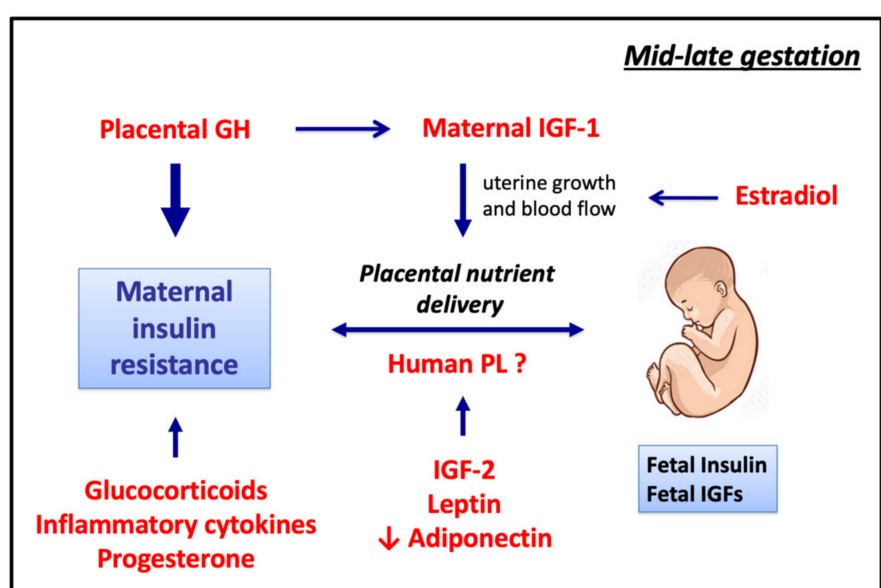

**Figure 5.** The roles of maternal and placental hormones in placental nutrient delivery and fetal growth. Adapted from Newbern D and Freemark M [47].

## 5. Control of Fetal Growth and Weight Gain by Placental and Fetal Hormones, Growth Factors, and Adipocytokines

The uptake and utilization of nutrients and the growth and weight gain of the fetus are controlled by placental and fetal hormones and growth factors; of these, the most important are insulin and the IGFs (Table 1). Other factors that may promote fetal growth and weight gain include thyroid hormone and sex steroids and, possibly, the lactogenic hormones placental lactogen and prolactin. Conversely, at high concentrations, glucocorticoids inhibit fetal and postnatal linear growth. Intrauterine growth restriction and fetal overgrowth are accompanied by adaptive changes in fetal levels of the adipocytokines leptin and adiponectin and the gastric hormone ghrelin, which regulate food intake, energy expenditure, insulin sensitivity, and GH secretion; whether or not the adipocytokines or ghrelin directly modulate fetal growth or weight gain is unclear.

**Table 1.** Control of human fetal growth and weight gain by placental and fetal hormones, growth factors, and adipocytokines.

| Fetal Hormone/Growth Factor/Cytokine (or Cognate Receptor) | Source | Proposed Biological Actions in the Fetus | Correlation of Fetal Level with Birth Length | Correlation of Fetal Level with Birth Weight/Fat Mass | Effect of Fetal Deficiency/Deletion or Excess (Human) on Fetal Weight or Length |
|---|---|---|---|---|---|
| **Insulin** | Yolk sac and fetal thymus in early development (*? contribution to fetal plasma levels*); then, pancreatic beta cells. | Exerts adipogenic, lipogenic, and glycogenic effects in fetal tissues and the induction of placental glucose transport and amino acid uptake. Stimulates amino acid uptake (but not DNA synthesis) in human fetal myoblasts and fibroblasts. | Weakly correlated with birth length. Low in SGA/IUGR. | Strongly correlated with birth weight and extremity and abdominal fat. | Marked decrease in birth weight, lesser decrease in birth length in pancreatic agenesis or defects in beta cell development or insulin production. Macrosomia in infants of diabetic mothers associated with fetal hyperinsulinemia. |
| **IGF-1** | Connective tissues and mesenchymal cells in diverse fetal tissues including perichondrium. Placental villous cytotrophoblast, extravillous trophoblast, and microvillous membrane. | Exerts mitogenic, anti-apoptotic, and adipogenic effects in fetal tissues. Induces placental trophoblast migration and invasion, and the differentiation of cytotrophoblasts into syncytiotrophoblasts; stimulates placental glucose transport and amino acid uptake. | Correlated with birth length and fat-free mass. Low in SGA/IUGR. | Strongly correlated with birth weight and fat mass. Increased in children born LGA. Low birth weight associated with low IGF-1 and high IGFBP-1 and IGFBP-2. | Severe IUGR in patients with mutations in IGF-1 or IGF-1 receptor. |
| **IGF-2** | Fetal endothelial cells, chondroblasts, hepatocytes, myocytes, adrenocortical cells, and epithelial cells of the developing kidney and choroid plexus. Placental chorionic plate, chorionic mesoderm, cytotrophoblast, and extravillous trophoblast. | Promotes trophoblast migration and invasion, placental growth and maturation, and nutrient transport. Exerts mitogenic, anti-apoptotic, and adipogenic effects in fetal tissues and promotes fetal chondogenesis and myogenesis. | Weakly corelated with birth length and fat-free mass. Low in SGA/IUGR. | Variably correlated with birth weight. Commonly low in SGA/IUGR; variably elevated in fetal macrosomia. | Severe IUGR in patients with paternally inherited IGF-2 missense mutations. Intrauterine growth restriction and placental hypoplasia in Russell–Silver syndrome associated with a loss of paternal expression of IGF-2; conversely, fetal overgrowth and placentomegaly in Beckwith–Wiedemann syndrome associated with bi-allelic IGF-2 (over) expression caused by a loss of maternal imprinting. |

**Table 1.** *Cont.*

| Fetal Hormone/Growth Factor/Cytokine (or Cognate Receptor) | Source | Proposed Biological Actions in the Fetus | Correlation of Fetal Level with Birth Length | Correlation of Fetal Level with Birth Weight/Fat Mass | Effect of Fetal Deficiency/Deletion or Excess (Human) on Fetal Weight or Length |
|---|---|---|---|---|---|
| **Pituitary GH (GH-N)** | Fetal pituitary | Effects on fetal IGF-1 blunted because of immaturity of hepatic GH receptor. Decline in levels after birth may facilitate insulin-dependent lipogenesis and fat deposition. | No significant correlation | None or inverse | Variable and mild decrease in birth length (mean −0.7–1.0 SD), relative adiposity (−0.3–0.4 SD) in hypopituitarism, GH deficiency, or mutations in GH receptor. |
| **Placental GH (GH-V)** | Placenta | Stimulates a rise in maternal IGF-1 after mid-gestation and may promote trophoblast invasion. Induces maternal insulin resistance in mid–late gestation. Little or no GH-V in the fetal circulation. | N/A | N/A. Low *maternal* levels in IUGR | Variable reduction in birth weight in GH-V gene deletions (in association with HPL deletions). |
| **Human Placental Lactogen (hPL)** | Placenta | Stimulates amino acid uptake, DNA synthesis, and IGF-1 secretion in human fetal myoblasts and fibroblasts and DNA synthesis and IGF-1 secretion in human fetal hepatocytes. Effects on human fetal or maternal pancreatic beta cell mass and insulin production unclear. | Fetal hPL correlates positively with fetal IGF-1 and IGF-2 in late gestation. Birth weight and length correlate positively with maternal but not cord blood hPL. | Fetal hPL correlates positively with fetal IGF-1 and IGF-2 in late gestation. Birth weight and length correlate positively with maternal but not cord blood hPL. | Variable reduction in birth weight in HPL gene deletions (some in association with GH-V deletions). |
| **Prolactin** | Fetal pituitary | Stimulates white and brown adipogenesis in rodent cell lines. May promote postnatal "whitening" of brown adipose tissue. Stimulates pancreatic beta cell replication in postnatal rodents but effects on human fetal or maternal pancreatic beta cell mass and insulin production unclear. | ? | Variably high in SGA infants | Decreased beta cell mass and brown adipose depots in neonatal PRLR-knockout mice. Combined defects in PRLR signaling and GH production in the mouse reduce weight on postnatal day 7. |

**Table 1.** *Cont.*

| Fetal Hormone/Growth Factor/Cytokine (or Cognate Receptor) | Source | Proposed Biological Actions in the Fetus | Correlation of Fetal Level with Birth Length | Correlation of Fetal Level with Birth Weight/Fat Mass | Effect of Fetal Deficiency/Deletion or Excess (Human) on Fetal Weight or Length |
|---|---|---|---|---|---|
| **Thyroid hormone** | Maternal and fetal thyroid | Increases BAT development and enzymes for thermogenesis in the neonatal period. Promotes fetal bone growth and bone maturation. | Weak association of cord free T4 (not free T3) with birth length. | Weak association of cord free T4 (not free T3) with birth weight. | Variable increase in birth weight and delayed bone maturation in severe primary congenital hypothyroidism; low birth weight in fetal hyperthyroidism. |
| **Cortisol** | Fetal adrenal | Promotes the maturation of respiratory and gastrointestinal functions and liver glycogenesis in late gestation. Modulates thyroid deiodinase activity. May facilitate expression of the hepatic GH receptor in late gestation and the early postnatal period. Inhibits linear growth at supraphysiologic concentrations. | Glucocorticoid excess (endogenous or exogenous) reduces IGF-1 and increases IGFBP-1 and inhibits linear growth. | Cord blood cortisol variably increased in IUGR; birth weight correlates with placental 11 beta-HSD2 activity. | Birth weight reduced markedly in patients with mutations in 11 beta-HSD2. Normal birth weight in congenital lipoid adrenal hyperplasia (mutation of Steroidogenic Acute Regulatory Protein). |
| **Sex steroids** | Placenta, fetal adrenal and gonads. | ? effects on fetal bone growth and maturation. | Birth length slightly greater in boys than girls. | Birth weight slightly greater in boys than girls. | Variable slight decrease in birth weight in males with complete androgen resistance; no effects of androgen synthetic defects, ovarian hypogonadism, or mutations in aromatase or estrogen receptor. Birth weight normal in states of androgen excess (congenital adrenal hyperplasia) in females or males. |
| **Leptin** | Placenta and fetal adipose | Increases sympathetic nervous system activity and fetal BAT development. Exerts growth-promoting effects in postnatal rodents. | Weak or no association with birth length. | Strongly correlated with birth weight and neonatal fat mass; low in intrauterine growth restriction. | Normal birth weight and length in patients with mutations in leptin or leptin receptor. |

**Table 1.** *Cont.*

| Fetal Hormone/Growth Factor/Cytokine (or Cognate Receptor) | Source | Proposed Biological Actions in the Fetus | Correlation of Fetal Level with Birth Length | Correlation of Fetal Level with Birth Weight/Fat Mass | Effect of Fetal Deficiency/Deletion or Excess (Human) on Fetal Weight or Length |
|---|---|---|---|---|---|
| **Ghrelin** | Placenta and fetal tissues including the pancreas. | Stimulates food intake, fat storage and growth hormone secretion in the postnatal period. | Inversely related to birth length (limited data). | Inversely related to birth weight z score. | Normal birth weight and length in a child with a mutation that abolishes constitutive ghrelin receptor activity. Birth weight normal in ghrelin knockout mice. |
| **Adiponectin** | Fetal vascular endothelial cells in skin, kidney, cortex, skeletal and smooth muscle. Low-level expression in fetal adipose. | Inhibits placental amino acid glucose transport and stimulates placental IGFBP-1 expression. Increases the expression of placental lactogen in human placental explants and cells. | ? | Rises markedly between 24 weeks and term and correlates variably with birth weight and adiposity. | Deletion in pregnant mice reduces placental IGFBP-1 expression and increases birth weight. |

**Abbreviations**: SGA, small for gestational age; IUGR, intrauterine growth restriction; LGA, large for gestational age; IGFBP-1, insulin-like growth factor binding protein 1; IGFBP-2, insulin-like growth factor binding protein 2; PRLR, prolactin receptor; 11 beta-HSD2, 11 beta hydroxysteroid dehydrogenase type 2.

Fetal hormones and growth factors are derived largely from production and secretion by fetal tissues and/or the placenta; there is little or no *trans*placental delivery of *maternal* insulin, IGFs, leptin, adiponectin, ghrelin, or the lactogens or GH to the fetus. On the other hand, placental transport of maternal thyroid hormones varies in degree and specificity during development. Certain glucocorticoids (Dexamethasone) also cross the placenta but maternal cortisol is largely inactivated by placental 11 beta hydroxysteroid dehydrogenase type 2 (11 beta-HSD 2).

## 6. Insulin

During early fetal development, insulin mRNA is detected in the yolk sac and fetal thymus [54,55]; whether or not these contribute to fetal insulin levels is currently unclear. Subsequently, the major source of insulin is the fetal pancreatic beta cell. In association with increases in fetal beta cell mass, the levels of fetal insulin and *c*-peptide rise from mid-gestation to term (Figure 6) [56,57].

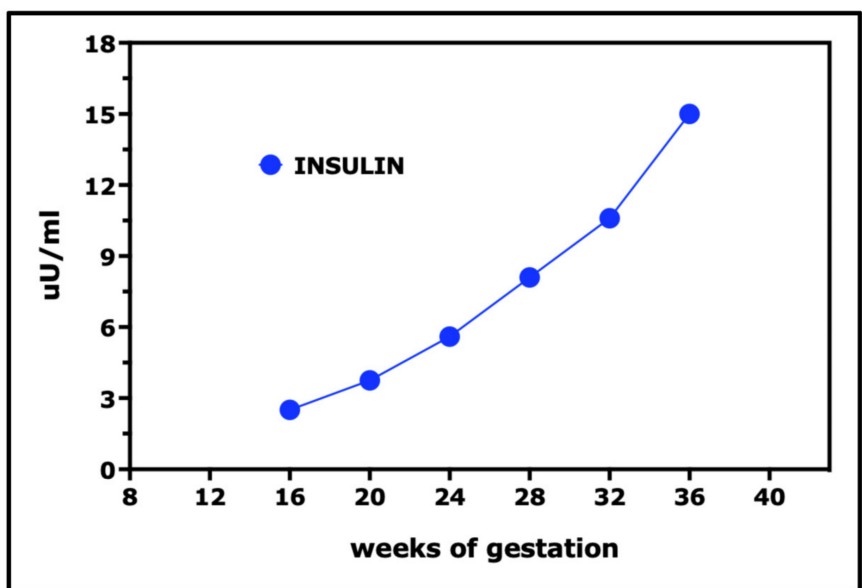

**Figure 6.** Serum insulin concentrations in the human fetus. The figure is adapted from cordocentesis data presented by Economides DL et al. [56] and Salardi S et al. [57].

Several lines of evidence establish a central role for insulin in the control of fetal weight gain and fat deposition. Insulin has potent adipogenic, lipogenic, and anti-lipolytic effects in white adipocytes [58] and increases placental glucose transport in early gestation and placental amino acid uptake in late pregnancy [59–62]. Likewise, insulin stimulates amino acid transport (but not DNA synthesis) in human fetal skin fibroblasts and myoblasts [63]. Accordingly, increases in fetal insulin levels parallel increases in body weight and fat storage during the second half of gestation. Cord blood insulin and *C*-peptide levels at term correlate strongly with birth weight and measurements of extremity and abdominal fat [64]. Fetal overgrowth (LGA) in infants of diabetic mothers is associated with fetal hyperinsulinemia [65]; conversely, *C*-peptide and insulin levels are low in children born small for gestational age (SGA), and there are marked decreases (~50%) in birth weight in pancreatic agenesis or loss of function mutations in the genes encoding insulin or the insulin receptor [66,67]. The rate of fetal weight gain in such cases declines markedly after 30 weeks' gestation, in accord with insulin effects on fetal fat deposition. Birth weight is more variable and less markedly reduced in infants with less severe defects in insulin production or signaling, including heterozygous mutations in glucokinase, HNF1alpha, HNF1beta, HNF4alpha, and various lipodystrophy genes [67].

Insulin induction of nutrient uptake and utilization in fetal tissues promotes body growth as well as weight gain. Yet the rise in fetal insulin during mid–late gestation

coincides with a decline in the rate of femur growth, and birth length is only mildly increased (~10%) in LGA infants of diabetic mothers and less dramatically reduced (−12%) than birth weight (−50%) in pancreatic agenesis or insulin gene mutations [66]. Relative to birth weight, percent fat mass is higher and percent lean body mass lower in infants of diabetic mothers [68,69]. Moreover, insulin had no effect on DNA synthesis in human fetal myoblasts or fibroblasts [63] but stimulated IGF-1 production in fetal rat hepatocytes [70]. Insulin reduces IGF binding protein 1 (IGFBP-1) levels in postnatal life and might thereby increase IGF bioactivity in the fetus (see below).

### 7. Insulin-like Growth Factors (IGFs) and IGF Binding Proteins

The insulin-like growth factors (IGFs) are the central mediators of linear growth in fetal as well as postnatal life, with mitogenic and anti-apoptotic effects in various tissues including growth plate chondrocytes. **IGF-1** mRNAs are detected in most fetal tissues; interestingly, the level of expression in placenta and fetal stomach, spleen, thymus, adrenal, muscle, and heart exceeds that in fetal liver [71]. The mRNAs localize to connective tissues and cells of mesenchymal origin including the perisinusoidal cells of liver and the perichondrium of cartilage [72]. In the placenta, IGF-1 is expressed in villous cytotrophoblast, extravillous trophoblast, and microvillous membranes; it promotes trophoblast migration and invasion and differentiation of cytotrophoblasts into syncytiotrophoblast and increases placental amino acid uptake and glucose transport [49].

IGF1 stimulates DNA synthesis and amino acid uptake in human fetal fibroblasts and myoblasts [63] and DNA synthesis in human fetal hepatocytes [73]. IGF-1 also promotes adipogenesis in white and brown preadipocytes [74–77] and glucose-stimulated insulin secretion in postnatal mice but appears to have no effect on fetal beta cell mass [78,79].

**IGF-2** is expressed at far higher levels than IGF-1 in the human placenta and diverse fetal tissues including the liver, adrenal gland, skin, and gonads [71,80]. IGF-2 mRNA is detected in fetal endothelial cells, chondroblasts, hepatocytes, myocytes, adrenocortical cells, and epithelial cells of the developing kidney and choroid plexus [81]. Like IGF-1, IGF-2 exerts potent mitogenic, anti-apoptotic, and adipogenic effects in human fetal and postnatal tissues and promotes chondrogenesis and myogenesis in human fetal mesenchymal cells [63,82,83]; these actions are thought to be mediated through binding to the IGF-1 and/or insulin receptors or to hybrid IGF-1/insulin receptors. Limited evidence in rodents suggests that IGF-2 may also promote beta cell replication during fetal and neonatal development [84,85]. Studies in human placenta in vitro and in sheep and transgenic mice in vivo demonstrate that fetal-derived IGF-2 promotes placental endothelial cell proliferation and survival, angiogenesis, and trophoblast morphogenesis and enhances the transplacental delivery of glucose and essential amino acids to the fetus [49,50,86]. The effects of IGF-2 on placental development and nutrient transport in the mouse may be mediated, at least in part, through its binding to the IGF2 receptor [50].

IGF-1 and IGF-2 are detected in human fetal blood at mid-gestation and rise progressively after 28–30 weeks to peak near term (Figure 7) [87,88]. Fetal plasma concentrations of IGF-2 exceed greatly the plasma concentrations of IGF-1. IGF-1 levels decline after birth following the loss of the placenta, while IGF-2 levels remain relatively stable.

In theory, the activity of the IGFs in early fetal development may be facilitated by the relatively low levels of IGF binding protein 3 (IGFBP-3) [87–90]. On the other hand, high levels of IGF binding protein 1 (IGFBP-1) (Figure 8) in stressed preterm infants [91,92] can inhibit IGF action.

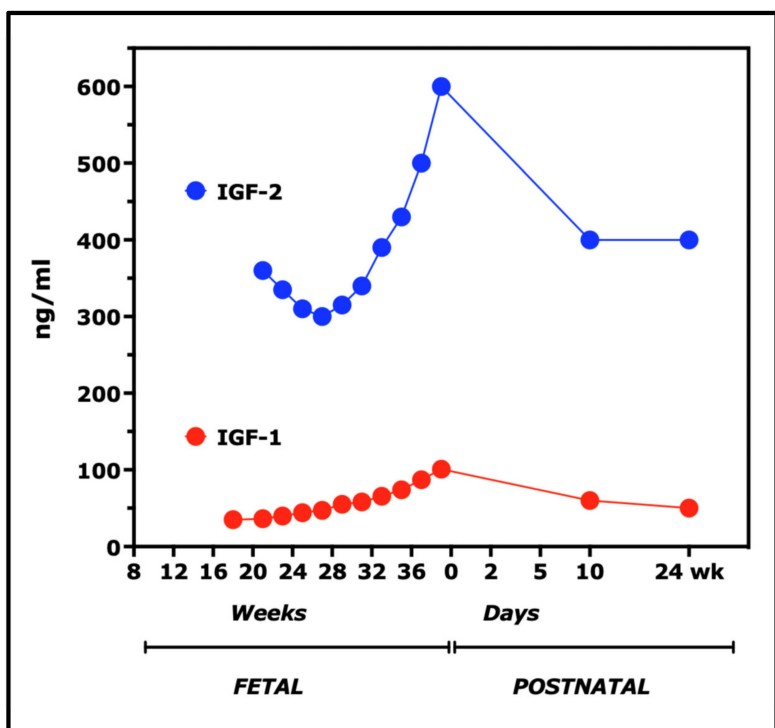

**Figure 7.** Serum insulin-like growth factor concentrations in the human fetus. The figure is adapted from cordocentesis data presented by Lassarre C et al. [87] and Langford K et al. [88]. IGF-1, insulin-like growth factor 1; IGF-2, insulin-like growth factor 2.

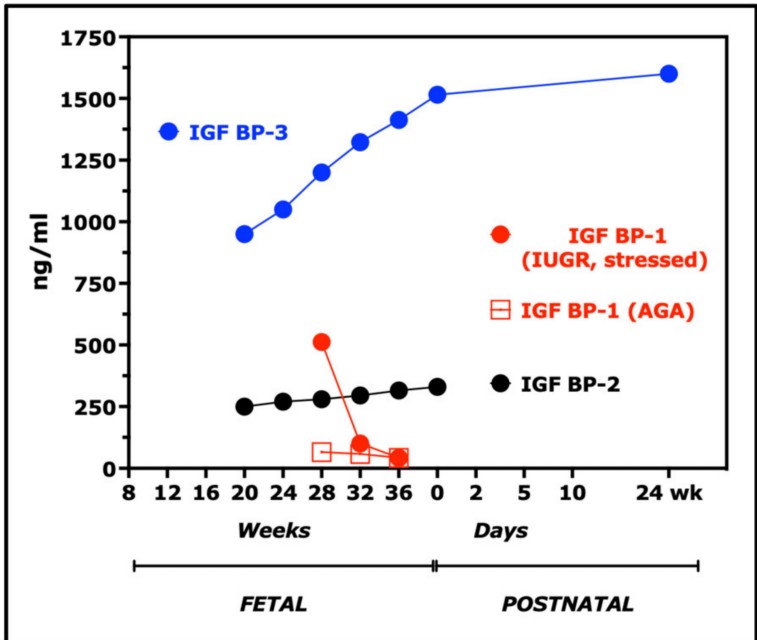

**Figure 8.** Serum concentrations of the insulin-like growth factor binding proteins in the human fetus. The figure is adapted from cordocentesis data presented by Langford K et al. [88] and Leger J et al. [93] and from cord blood data in Hellstrom A et al. [91]. IGFBP-1, insulin-like growth factor binding protein 1; IGFBP-2, insulin-like growth factor binding protein 2; IGFBP-3, insulin-like growth factor binding protein 3; IUGR, intrauterine growth restriction; AGA, appropriate for gestational age.

IGF-1 levels in cord blood correlate strongly with birth weight, fat mass, and (to a lesser extent) birth length [94–97]. Fetal macrosomia (LGA) is accompanied by increases in cord blood IGF-1 (and insulin) levels and a low level of IGFBP-1 [94,97]. Conversely, low

birth weight is associated with low cord IGF-1 and high IGFBP-1 and IGFBP-2 [92,94,98,99]. Children with loss of function mutations in the IGF-1 gene or haploinsufficiency of IGF-1 (terminal deletion of chromosome 15q) have severe IUGR (birth weight −2.5 to −4.5 SD; birth length −3.7 to −6.5 SD) and postnatal growth failure. Heterozygous mutations of the IGF-1 receptor also reduce birth weight (−1.5 to −3.5 SD) and length (−0.3 to −5.8 SD) [100–102].

IGF-2 levels in cord blood correlate with birth weight and length, though not as strongly or consistently as IGF-1. In most, but not all, studies, cord blood IGF-2 levels are low in children born small for gestational age; conversely, cord blood IGF-2 is variably elevated in macrosomic (LGA) infants [87,95–98,103,104] (see also discussion of maternal diabetes below). Simple correlations between circulating IGF-2 levels and fetal growth may be misleading, however, as much of the growth promoting effect of IGF-2 in the fetus may be mediated through autocrine and/or paracrine effects on placental growth and function [49,50,83,86].

IGF2 is an imprinted gene that is normally expressed only by the paternal allele in the placenta and fetal tissues other than the brain [105]. Intrauterine growth restriction and placental hypoplasia in Russell–Silver syndrome are associated with an absence of paternal IGF2 expression, resulting from a loss of methylation of the distal imprinting control region (H19/IGF2:IG-DMR) on 11p15.5 or IGF2 point mutations [83,106]. Conversely, fetal overgrowth and placentomegaly in Beckwith–Wiedemann syndrome are associated with bi-allelic IGF2 (over) expression resulting from mosaic segmental paternal uniparental isodisomy of 11p15.5 or gain of methylation at the maternal H19/IGF2:IG-DMR allele [83]. Severe IUGR was first noted in a family with a paternally inherited IGF2 mutation [106]; a recent report identified five additional missense mutations of IGF2 associated with striking fetal growth restriction [107]. Circulating IGF-2 levels are normal after birth in children with methylation defects but are low in those with point mutations; interestingly, the latter have elevated levels of IGF-1 [107].

## 8. Regulation of Fetal Insulin, IGFs, and IGF Binding Proteins by Nutrients and Hormones

### 8.1. Macronutrients

Fasting and prolonged nutrient deprivation during postnatal life reduce plasma insulin, leptin, and IGF-1 levels and increase plasma cortisol, ghrelin, and growth hormone [108]. Reductions in IGF-1 in fasted and malnourished children and adults result from the down-regulation of the hepatic GH receptor and post-receptor defects in growth hormone (GH)-STAT5 signaling [109–111] and are particularly sensitive to protein restriction. The effects of fasting are reversed by refeeding: in combination with protein, the intake of carbohydrates may increase IGF-1 more than fats. Effects of nutrient deprivation on IGF-1 and growth in postnatal life are thought to be mediated in part through induction of FGF21, which down-regulates GH receptor expression and inhibits GH-dependent phosphorylation of STAT5 [112].

Macronutrients provide the essential substrates for fetal as well as postnatal weight gain and linear growth. Placental transport of glucose, amino acids, and essential fatty acids to the fetus increases progressively during mid–late gestation in response to a rise in maternal nutrient availability together with increases in placental size, vascularity, and expression of GLUT1 and System A (neutral) amino acid transporters [86,113,114] and placental lipases [115]. Macronutrients modulate fetal weight gain and growth through effects on fetal insulin, IGF-1, and the expression of IGF binding proteins. Insulin secretion in the ovine fetus is induced by glucose and amino acids [116,117], and hyperglycemia in macrosomic infants of diabetic mothers is accompanied by elevations in fetal insulin and, variably, IGF-1 [118,119] (see discussion of Gestational Diabetes below). Conversely prolonged fasting of pregnant rodents and sheep reduces maternal and fetal insulin and IGF-1 levels [120–122] and increases fetal IGFBP-1 [116].

Whether or not macronutrients exert direct control over fetal IGF production is less clear; the effects of macronutrients on IGF expression in fetal tissues are inconsistent, exerted at high concentrations, and difficult to distinguish from general effects on cellular health and nutritional balance. For example, glucose up-regulates IGF-1 and IGF-2 mRNA in fetal rat hepatocytes at concentrations of 10–20 mM [70]; human umbilical cord glucose levels at term normally range from 3.5 to 4.5 mM. In one study [121], a glucose infusion restored IGF-1 levels in fetal sheep subjected to prolonged maternal fasting; the infusion of amino acids had no effect. Another study found no effect of glucose infusion on IGF-1 or IGF-2 in the fasted ovine fetus [122]. A role for FGF21 in fetal insulin or IGF production seems less likely because FGF21 was not detected in the cord blood of the great majority of healthy term infants [112].

IGF action in the fetus is modulated by macronutrient effects on IGF binding protein expression. Particularly responsive is IGFBP-1, which inhibits IGF action in fetal and postnatal tissues [123]. Hepatic IGFBP-1 mRNA levels and IGFBP-1 plasma concentrations in the ovine fetus are increased by maternal fasting and fetal hypoglycemia and reduced by maternal and fetal hyperglycemia [116]. Likewise, cord blood IGFBP-1 levels in infants of diabetic mothers correlate negatively with birth weight [119]. Nutritional effects on IGFBP-1 expression are likely mediated through changes in fetal insulin levels [116].

### 8.2. Growth Hormone (GH), Placental Lactogen, and Prolactin

The major hormone controlling IGF-1 production in postnatal life is pituitary GH. However, levels of fetal IGF-1 correlate inversely with **fetal pituitary GH** levels (Figure 9) [57,93]; the relative insensitivity of fetal IGF-1 and fetal growth to GH is thought to reflect the immaturity of the hepatic GH receptor [124–126], which manifests as high levels of GH [93,127–129] and low levels of circulating growth hormone binding protein (GHBP) in the fetus at mid-gestation and term and a progressive increase in GHBP levels after 3–6 months of age [130–132]. This likely explains the variable and mild reductions in birth length (mean −0.7–1.0 SD) and lesser reductions in birth weight (mean −0.3–0.4 SD) of children with GH receptor mutations and the normal or low normal birth weight and length in infants with GH gene mutations, PROP1 mutations, congenital GH deficiency, and mutations in STAT5B, a nuclear transducer of GH signaling [100–102,133]. Clinically significant growth failure in children with defects in GH production or signaling generally emerges after 3–6 months of (postnatal) age [134].

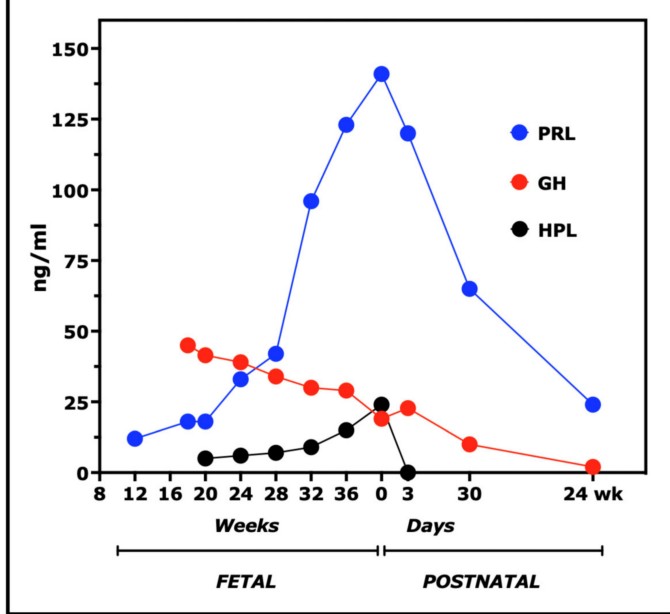

**Figure 9.** Serum concentrations of growth hormone (GH), placental lactogen (HPL) and prolactin (PRL) in the human fetus and newborn infant. The figure is adapted from cordocentesis data in

Leger J et al. [93] and Salardi S et al. [57] (GH, growth hormone); Lassarre C et al. [87] (hPL, human placental lactogen); and Arosio M et al. [128] and Thorpe-Beeston JG et al. [135] (PRL, prolactin). Postnatal data are adapted from Leger J et al. [127] and Binder G et al. [129] (GH) and Perlman M et al. [136] (PRL).

GH promotes adipogenesis and IGF-1 expression in mouse preadipocytes [137] but the effects of GH on development of human fetal or neonatal adipose tissue have not yet been characterized. Nevertheless, newborn infants with GH deficiency are predisposed to hypoglycemia, presumably as a consequence of increased sensitivity to insulin. It follows that the rapid decline in GH levels in the full-term infant after 10 days of life [127,129] may facilitate insulin-dependent lipogenesis and postnatal fat deposition.

In cordocentesis samples the levels of **human placental lactogen (hPL)** in the fetal circulation (Figure 9) rise after 20 weeks' gestation [87]. In vitro studies performed more than 30 years ago showed that hPL stimulates IGF-1 production, DNA synthesis, and amino acid uptake in human fetal skin fibroblasts and myoblasts [63,138] and IGF-1 production and DNA synthesis in human fetal hepatocytes [73]. A 14-day infusion of ovine PL in the sheep fetus increased fetal IGF-1 levels by 42% [139]. The relationships between human fetal/cord blood hPL levels and birth weight and length have not been studied in depth; most investigations have examined the relation between maternal hPL levels and fetal growth. The best study [87] found that fetal hPL correlated positively with fetal IGF-1 and IGF-2 levels after 33 weeks' gestation but did not correlate with ultrasound estimates of fetal weight. Likewise, birth weight and length at term correlated positively with maternal but not cord blood hPL levels [140,141]. On the other hand, Hill et al. [142] showed that the weight of human fetuses at 12–19 weeks' gestation correlated positively with hPL binding capacity of fetal liver. Genetic deletions of HPL, some in combination with deletion of GH-V (placental GH), are accompanied by IUGR in some but not all cases [47]. Interestingly, a targeted knockdown of the ovine PL gene caused variable reductions in placental and fetal weight and crown-rump length with reductions in uterine blood flow, fetal liver IGF-1 and IGF-2 mRNAs, and umbilical artery IGF-1 concentrations [143,144]. The implications of these findings for human physiology are unclear, as ovine PL has potent somatogenic as well as lactogenic activity, while hPL and prolactin bind the prolactin but not the GH receptor with high affinity [47].

In contrast to hPL, **placental GH** is in most studies not detected in fetal blood; one investigation, however, found vanishingly low levels (median 0.13 ng/mL) in umbilical cord blood at term [145]. Its effects on fetal growth appear to be mediated indirectly through modulation of maternal insulin action and nutrient availability [47,146]. However, placental GH also promotes trophoblast invasion in early gestation and may increase uterine blood flow [47,146]. Levels of placental GH in the mother (but not fetus) correlate with birth weight and are reduced in SGA infants [47,146].

**Prolactin** rises to high levels in the late gestational human fetus and newborn infant (Figure 9) [128,135,136], likely reflecting the induction of lactotrope proliferation by rising estrogen levels [147]. The relative hyperprolactinemia in the early postnatal period lasts longer in preterm than in term infants [136]. Yet its contributions to fetal and neonatal growth and weight gain are poorly understood. A role for prolactin in fetal and neonatal adipose development is suggested by four lines of evidence: (a) the rise in prolactin in the fetus during the latter half of gestation and the high levels of prolactin in the neonatal period coincide with striking increases in white and brown fat stores; (b) prolactin stimulates adipogenesis in mouse preadipocytes [137,148] and may promote the differentiation of brown adipocytes [149] and their "whitening" after birth [149,150]; (c) prolactin signaling can increase fat deposition in rodents [150,151] and, in combination with hypogonadism, may promote weight gain in adult and adolescent patients with prolactinomas or medication-induced hyperprolactinemia [152–154]; and (d) white and brown adipose tissue mass and adipocyte number are reduced in prolactin receptor (PRLR) knockout mice [155,156]. In

contrast, IGF-1 levels are normal in adults and children with hyperprolactinemia (without acromegaly) and in prolactin receptor-deficient mice [157].

Prolactin and placental lactogen stimulate beta cell proliferation and glucose-stimulated insulin secretion and inhibit beta cell apoptosis in rodent islets in vivo and in vitro [158,159] and in some studies of human islets [160,161]. Moreover, lactogen signaling is required for pregnancy-dependent increases in beta cell mass in the mouse and maintenance of maternal glucose tolerance [162–164]. The effects of prolactin and placental lactogen are mediated through binding to prolactin receptors (PRLRs), which are detected in a wide array of human fetal tissues [165] including the liver, lung, duodenal villi, adrenocortical cells, renal tubular epithelial cells, skeletal myocytes, and chondrocytes of developing bones. Interestingly, a study of receptor expression in the human fetus at 7.5–14 weeks' gestation found PRLR immunoreactivity in pancreatic exocrine tissue and ducts but not in insulin-secreting beta cells or glucagon-secreting alpha cells [165]. A subsequent investigation in middle aged adults identified PRLRs in alpha and pancreatic polypeptide cells but not in pancreatic beta cells [166]. Whether or not PRLRs are expressed in human beta cells during pregnancy or later stages of fetal development is unknown; it may be relevant that prolactin, growth hormone, and estrogen, which circulate at high levels in the human mother and fetus during pregnancy, can up-regulate PRLRs in rodent beta cells [167]. In any case, the roles of the lactogens in beta cell development and insulin secretion in the human fetus and pregnant mother remain unclear. Some studies suggest that pituitary GH, through binding to GH receptors, might function as a human fetal or neonatal beta cell tropin [168–170].

Levels of prolactin are increased in SGA fetuses at 24–30 weeks of gestation [135] and in some but not all studies of SGA infants at term [171,172]. The effects of deletion of PRL or the PRLR on human birth weight or length have not yet been described. In mice, neither PRLR deletion nor GH deficiency had an effect on body weight on postnatal day 7. However, combined defects in PRLR signaling and GH production reduced weight on day 7 by 11% [157]. Interestingly, the males with combined defects developed adiposity and insulin resistance as they aged. Such combined defects in PRL and GH signaling have not been identified in humans. However, children and adults with STAT5B mutations are hyperprolactinemic [133] as well as GH-resistant. Since STAT5B is responsive to prolactin in target tissues [137,173], patients with STAT5B mutations likely have a *relative* resistance to lactogen as well as GH signaling. As noted previously, infants with STAT5B mutations have normal or low normal birth weight and length [133,174,175].

## 9. Thyroid Hormones

Thyroid hormone concentrations in fetal blood are modulated by the transplacental transport of thyroid hormones from the mother to fetus, TSH-dependent thyroid hormone production by the fetal thyroid, tissue dependent activation of thyroid hormones by types 1 and 2 deiodinases, and inactivation by type 3 deiodinase and T3 sulfotransferases [176]. In early and mid-gestation, fetal T4 is sustained by maternal fetal transport [177,178]; low T3 and high reverse T3 levels reflect low levels of type 1 deiodinase (which converts T4 to T3) and high-level expression of type 3 deiodinase (which converts T4 to the inactive reverse T3) and thyroid hormone sulfotransferases in the placenta and fetal tissues [179–181]. Accordingly, thyroxine administration to infants <30 weeks' gestation raises free T4 and reverse T3 but not total T3 levels [182]. Fetal T4 levels rise progressively after 16–20 weeks of gestation in response to increasing production of TSH by the fetal pituitary gland (Figure 10) [183–185]. T3 levels rise and reverse T3 declines after 30 weeks' gestation, with cortisol- and (possibly) leptin-dependent increases in type 1 deiodinase activity and a decrease in type 3 deiodinase [186–188]. Loss of the placenta at delivery contributes to a postnatal surge in T3.

Bone maturation is commonly delayed in infants with congenital hypothyroidism, and thyroid hormones promote growth and maturation of long bones in the ovine fetus in late gestation [189]. Thyroid hormones increase expression of GH in the fetal pituitary

but IGF-1 levels are normal in thyroidectomized fetal sheep [190]. There are relatively weak positive correlations between cord blood free T4 (but not free T3 or TSH) and birth weight (r = 0.25) and length (r = 0.17) [191] in the human fetus; however, birth weight is variably *increased* in children with severe congenital hypothyroidism, in some cases because of post-term birth. Conversely, fetal hyperthyroidism reduces weight gain in utero but accelerates fetal bone maturation [192].

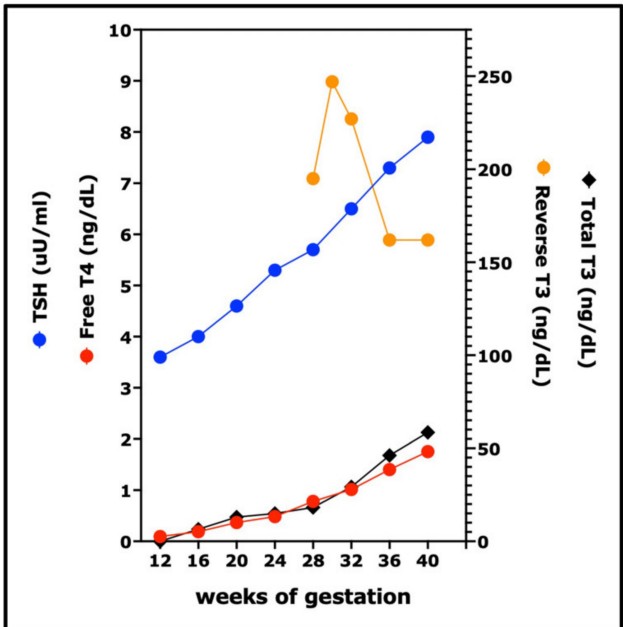

**Figure 10.** Serum concentrations of thyroid hormones and thyroid stimulating hormone (TSH) in the human fetus. The figure is adapted from cordocentesis data in Thorpe-Beeston JG et al. [183] and Radunovic N et al. [184] and from cord blood data in Santini F et al. [180].

Thyroid hormones increase UCP-1 expression and thermogenic activity in ovine fetal brown adipose tissue [193]; fetal hypothyroidism is accompanied by "whitening" of brown adipose stores and a decrease in whole body oxygen consumption [194]. The thyroid hormones also promote maturation of the fetal hypothalamic pituitary adrenal axis; thyroidectomy of the ovine fetus attenuates the surge in ACTH and cortisol in late gestation [195].

## 10. Cortisol and the Sex Steroids

**Cortisol** levels in the human fetus normally rise in late gestation in parallel with a decline in placental 11 beta-HSD 2 activity and activation of the fetal hypothalamic–pituitary– adrenal axis [196,197]. The late-gestational surge in cortisol increases liver glycogen stores and induces expression of gluconeogenic enzymes to ensure maintenance of blood glucose following loss of the placenta [198]. In the ovine fetus, cortisol promotes expression of digestive enzymes and type 1 deiodinase and reduces type 3 deiodinase in placenta and fetal tissues, thereby increasing fetal T3 concentrations [186]. The surge in cortisol in the ovine fetus increases the expression of the hepatic GH receptor and IGF-1, prefiguring the change from GH-independent fetal growth to GH-dependent postnatal growth [124,198].

Umbilical cord cortisol levels correlate inversely with IGF-1 levels and positively with IGFBP-1 [91] and are elevated (variably) in small-for-gestational-age infants [199,200], particularly before 33 weeks of gestation [201]. This likely represents an adaptive response to intrauterine stress and nutrient deprivation. Nevertheless, exposure of the fetus to supraphysiologic concentrations of glucocorticoids impedes fetal growth: birth weight is reduced markedly in patients with inactivating mutations in 11 beta-HSD2 [202], which converts cortisol to the inactive cortisone. 11 beta-HSD2 is expressed in the placenta and various fetal tissues, and down-regulation of placental 11 beta-HSD2 activity is associated

with intrauterine growth restriction [203]. On the other hand, children with congenital lipoid adrenal hyperplasia (mutation of Steroidogenic Acute Regulatory Protein), who have deficiencies of cortisol and mineralocorticoids, have normal birth weight [204].

**Sex steroids**: Mean birth weight (3.35 kg) and length (49.89 cm) in healthy term boys exceed mean birth weight (3.23 kg) and length (49.15 cm) in healthy term girls by ~3.5% and 1.5%, respectively (WHO growth standards). Sex differences in fetal weight may be more pronounced in mid- than in late gestation [205]. The roles of sex steroids in fetal weight gain and growth remain unclear; one study [206] found that chromosomal males with androgen receptor mutations and complete androgen insensitivity have birth weights comparable to chromosomal females. Two other investigations, however, showed normal birth weights in chromosomal males with complete androgen resistance [207,208]. Likewise, birth weight was normal in chromosomal males with androgen synthetic defects [208] and in chromosomal males and females with androgen excess due to congenital adrenal hyperplasia [207,208]. Birth weight was also normal in the few studied cases of chromosomal girls with non-syndromic gonadal insufficiency [208] and males with mutations in aromatase [209] or the estrogen receptor [210]; these observations mitigate against an effect of estrogen on fetal weight gain. It should be noted that birth lengths in these various disorders have not been well characterized.

## 11. Leptin, Ghrelin, Adiponectin, and Other Adipocytokines

Adipocytes produce cytokines (adipocytokines) that modulate food intake, weight gain, fat deposition, and intermediary metabolism. Adipocytokines are also expressed by the human placenta as well as stromal and vascular endothelial cells. The cytokine **leptin** inhibits food intake, enhances satiety, promotes brown adipose tissue development and thermogenesis [211,212], and increases sympathetic nervous system activity and the conversion of T4 to the more active T3. In human placental villous fragments, leptin may promote the uptake of neutral amino acids [51,52,61].

Leptin expression by the placenta declines during pregnancy, with an up-regulation of leptin production by maternal and fetal adipose tissues [213–215]; the levels of leptin in fetal blood rise during late gestation (Figure 11) [216,217], in association with striking increases in fetal fat mass. Accordingly, leptin levels in umbilical cord at term correlate positively with birthweight and neonatal adiposity and are low in children with intrauterine growth restriction [218–220].

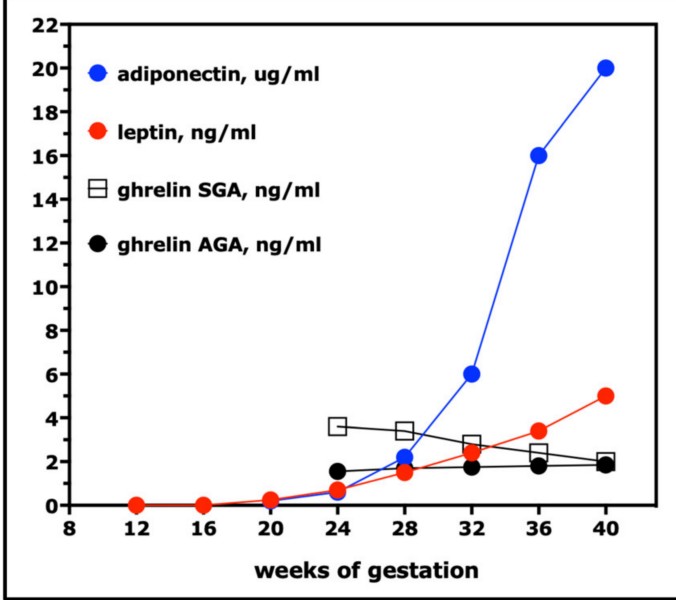

**Figure 11.** Serum concentrations of adipocytokines and ghrelin in the human fetus. The figure is adapted from cordocentesis data in Cetin I et al. [216] and Geary M et al. [217] (leptin) and cord blood

data in Kajantie E et al. [221] (adiponectin) and Kitamura S et al. [222] and Han L et al. [223] (ghrelin). SGA, small for gestational age; AGA, appropriate for gestational age.

Leptin stimulates bone growth in postnatal rodents [224] and increases circulating IGF-1 levels in children with leptin deficiency [225]. However, four considerations mitigate against a direct role of leptin in fetal and early postnatal growth: (1) leptin levels are low or undetectable at mid-gestation when the rate of long bone growth peaks; (2) leptin levels increase progressively during late gestation when the rate of bone growth declines; (3) birth length and weight are normal in children with mutations in leptin or the leptin receptor [225,226]; and (4) the rate of linear growth is normal in infants and young children with defects in leptin production or signaling [225,226].

In contrast to leptin, the hormone **ghrelin** promotes food intake and fat deposition in postnatal life; through the induction of GH secretion and suppression of insulin [227,228], it maintains euglycemia during periods of nutritional deprivation [108]. In humans, ghrelin is secreted by the first trimester placenta and by various fetal tissues [229,230]. Its role in fetal development remains unclear, as ~90% of the ghrelin in cord blood at term is by report unacylated (less active) [231]. Nevertheless, both total (better documented) and acylated ghrelin levels are higher in SGA than AGA preterm and term infants (Figure 11) and correlate inversely with birth weight, fat mass and, in some cases, birth length [222,223,232–234]. The rise in ghrelin may be an adaptive response to, rather than a cause of, fetal nutrient deprivation because birth weight is normal in ghrelin knockout mice [235] and weight and length were normal at birth (34.5 weeks) in a child with a compound heterozygous mutation that abolishes constitutive signaling through the ghrelin receptor [236].

**Adiponectin**, an insulin-sensitizing adipokine, is produced by vascular endothelial cells in human fetal skin, kidney, cortex, and skeletal and smooth muscles. Adiponectin mRNA and immunoreactivity are also detected in human fetal subcutaneous (white) adipose at term; however, levels of expression in the fetus are far lower than those in maternal white adipose tissue [237]. One study detected low levels of adiponectin immunoreactivity in brown adipose tissue in early gestation [238]. In contrast to levels in the pregnant mother, the levels of adiponectin in the fetal circulation rise markedly after 24 weeks' gestation [221] to a peak at term (Figure 11). The active, high molecular weight form predominates [237]. Thus, infants born prior to term have a relative adiponectin deficiency [239].

Though its effects on fetal tissues have not yet been characterized, adiponectin stimulates secretion of placental lactogen in human placental explants and cells [240] and reduces expression of glucose and amino acid transporters in first trimester placental trophoblasts [53]. By implication, the fall in adiponectin in the mother during mid–late gestation may up-regulate placental nutrient transporters and thereby promote fetal growth. Accordingly, maternal adiponectin levels correlate inversely with birth weight [241,242]. On the other hand, cord blood levels of adiponectin correlate positively with birth weight in some [99,243] but not other [244] studies. Cord adiponectin levels are mildly reduced in small-for-gestational-age infants at term [99]; however, those with low birth length but normal birth weight ("stunted") have normal adiponectin [99]. Adiponectin deletions in pregnant mice reduce placental IGFBP-1 expression; together with possible increases in placental nutrient transport [53,245], this may explain the increase in birth weight of heterozygous offspring.

Thus far, the roles of other adipocytokines in the control of metabolism and growth in the human fetus and newborn infant are poorly characterized. The insulin sensitizer **apelin** is expressed by human placenta and various fetal tissues and promotes placental amino acid uptake and glucose transport [246,247]. Limited evidence suggests that cord blood apelin levels are reduced in pregnancies complicated by maternal obesity [248].

The pro-inflammatory peptide **resistin** is expressed by cord blood mononuclear cells and, to a lesser extent, the placenta [249]. Resistin levels are higher in preterm than in term infants and exceed those in the pregnant mother. Most studies thus far find no clear relationship between cord blood resistin and metrics of fetal growth or weight

gain [250,251]; one investigation found variably higher levels in infants of mothers with gestational diabetes [249].

**Visfatin** is expressed at high levels in visceral adipose tissue and regulates tissue inflammatory responses, cell adhesion, vascular remodeling, and atherogenesis. Circulating levels are high in adults and children with obesity, metabolic syndrome, and type 2 diabetes [252,253]. Yet a small study found higher levels of visfatin in cord blood of SGA girls but not boys [254].

Finally, one investigation found high levels of **fetuin-A** in macrosomic infants of women with gestational diabetes [255].

## 12. Effects of Maternal Disorders on Fetal Hormones, Growth Factors, and Adipocytokines and Fetal Growth

### 12.1. Obesity

Maternal obesity, affecting nearly 30% of pregnant women in the United States [256], increases the risks of pregnancy complications including fetal macrosomia [257]. Obesity during pregnancy is defined as pre-pregnancy body mass index (BMI) $\geq 30$ kg/m$^2$. Differences in growth in fetuses of women with obesity compared to those without obesity appear as early as the second trimester [258]. By 21 weeks of gestation, femur and humerus lengths are significantly greater (Table 2) for fetuses of women with obesity (BMI $\geq 30$ kg/m$^2$) compared to women without obesity (BMI 19–29.9 kg/m$^2$). Fetal abdominal circumference is also greater when compared to fetuses of normal-weight women (BMI 19–24.9 kg/m$^2$). The differences in femur and humerus length and abdominal circumference persist through 38 weeks [258]. Beginning at 30 weeks of gestation, estimated fetal weight is significantly higher for the fetuses of women with obesity; weight curves diverge progressively from normal with increasing gestational age [258]. Head circumference is also larger on average in the fetuses of women with obesity, most significantly between 33 and 35 weeks. However, biparietal diameter is unchanged [258].

**Table 2.** Fetal growth trajectories in various maternal conditions affecting pregnancy.

| Maternal Condition | Second Trimester | Third Trimester | Fullterm Birth |
|---|---|---|---|
| **Obesity** | ↑ femur and humerus lengths (~21 weeks) | ↑ femur and humerus lengths<br>↑ EFW (~30 weeks)<br>↑ AC (~32 weeks) | ↑ length<br>↑ weight<br>↑ risk of LGA |
| **Gestational diabetes** | Possible period of ↓ EFW and AC (~14–20 weeks)<br>↑ femur length (~24 weeks)<br>↑ AC (~24 weeks) | Rapid growth<br>↑ femur length<br>↑ AC<br>↑ subscapular thickness<br>↑ fat and lean mass of the upper arm and thigh<br>(~28 weeks) | Larger and heavier than peers with:<br>↑ total tissue mass<br>↑ percentage fat mass<br>↑ risk of LGA |
| **Smoking** | ↓ femur length (~18–24 weeks)<br>↓ HC (~25 weeks) | ↓ femur length<br>↓ HC<br>↓ AC<br>↓ EFW | ↑ ponderal index<br>↓ length > ↓ HC > ↓ weight<br>↑ risk of SGA |
| **Severe preeclampsia** | ↓ EFW (~22 weeks)<br>↓ AC (~23 weeks) | ↓ AC<br>↓ EFW | *Likely to be delivered pre-term*<br>↓ weight > ↓ ponderal index > ↓ length<br>↑ risk SGA and asymmetric SGA |

Abbreviation: AC, abdominal circumference; EFW, estimated fetal weight; HC, head circumference; LGA, large for gestational age; SGA, small for gestational age; weeks, weeks gestational age.

Cord blood leptin and *c*-peptide levels are elevated in infants born to mothers with obesity (Table 3) compared to mothers with normal BMI [259–262]. The impact of maternal obesity on cord blood adiponectin is less clear, with some studies reporting no differences [259,261] and others reporting increased cord blood adiponectin [260]. Cord blood

ghrelin levels were not significantly different in pregnancies with obesity compared with those in normal weight women [261]. Similarly, there was no association between maternal obesity and cord blood cortisol [263] or IGF-1 levels [264,265].

**Table 3.** Effect of maternal conditions on cord blood hormone levels.

| Maternal Condition | Cord Blood Hormone Levels | | | | | | |
|---|---|---|---|---|---|---|---|
| | IGF-1 | IGFBP-3 | Leptin | Adiponectin | Ghrelin | Insulin/ *C*-Peptide | T4/fT4 |
| **Obesity** | ↔ | - | ↑ | ↔ | ↔ | ↑ | - |
| **Excessive gestational weight gain** | ↔ | ↑ | ↑ | ↔ | ↑ (? Males only) | ↑ | - |
| **Gestational diabetes** | ? (? Sex-dependent) | - | ↑ | ? | ? | ↑ | ↔ |
| **Smoking** | ↓ | ? | ↔ | ? | ↑ | ↔ | - |
| **Hypertensive Disorders of Pregnancy** | ↓ | - | ? | ↑ | ? | - | ↓ |
| **Malnutrition** | ↑ ? | - | - | - | - | - | ↓ ? |

↑, increased; ↔, no difference; ↓, decreased; ?, data variable or conflicting; -, unknown/unstudied.

### 12.2. Excessive Gestational Weight Gain

Like maternal obesity, gestational weight gain that exceeds the standards recommended by the Institute of Medicine is associated with pregnancy complications including fetal macrosomia [266]. In the United States, nearly half of pregnancies meet criteria for excessive gestational weight gain [267]. Excessive gestational weight gain was associated with increased cord blood levels of IGFBP-3, *c*-peptide, and leptin [268,269], while cord blood levels of adiponectin, IGF-1, and IGF-2 were normal [264,268], Table 3. One study found that infants born to women with excessive gestational weight gain had increased cord blood levels of total ghrelin as well as leptin. However, when each sex was considered separately, the levels were increased significantly only in the male infants. Moreover, cord blood ghrelin correlated negatively with birth weight and length [270].

### 12.3. Gestational Diabetes

The prevalence of gestational diabetes mellitus worldwide varies from 6 to 13% depending on geographic location [271]. Gestational diabetes increases the risks of maternal and neonatal complications, including macrosomia [272]. Screening for gestational diabetes is generally recommended between 24 and 28 weeks gestational age, although its impacts on fetal growth may begin earlier [272].

Studies of the effects of gestational diabetes on fetal growth in early pregnancy are conflicting [273–276], with some showing decreased estimated fetal weight and abdominal circumference between 14 and 24 weeks of gestation [275–277] and others finding no significant differences in abdominal circumference at 20 weeks [273,274]. Beyond 24 weeks, the fetuses of mothers with gestational diabetes have increased femoral diaphysis length and abdominal circumference (Table 2), particularly during the final weeks of gestation [274]. There is also greater abdominal fat thickness, subscapular fat thickness, and thigh and upper arm lean and fat mass, which appear by about 28 weeks of gestation and become more pronounced with increasing gestational age [274]. Overall fetuses of mothers with gestational diabetes have greater total tissue mass and higher ratio of fat mass to lean mass [274]. While larger head circumference can be seen around 24 weeks of gestation in fetuses of women with gestational diabetes, this difference is typically no longer significant at term [276]. At birth, the ratio of head circumference to abdominal circumference may be decreased [273,277].

Very little is known about the impacts of gestational diabetes on fetal hormones prior to delivery. Increasing maternal glucose levels are associated with increased cord blood *c*-peptide levels (Table 3), even if the criteria for gestational diabetes are not met [278–281]. Similarly, among pregnant women with or without gestational diabetes, cord blood leptin levels rise in association with increasing maternal fasting blood sugar levels [262,279–281]. Cord blood leptin and leptin per kg of infant weight are higher in newborns of mothers with gestational diabetes compared to normal pregnancies [216]. The higher leptin levels correlate with increased fetal abdominal fat mass [216].

The effect of gestational diabetes on cord blood ghrelin and adiponectin varies by study, with some finding lower levels of cord ghrelin and adiponectin compared to those in pregnancies of normoglycemic women [282,283] and others finding no significant differences [284–286] One study noted a decrease in the ratio of adiponectin:leptin in cord blood [262]. Cord blood thyroid function studies generally fall in the normal range [287].

Several studies reported that cord blood IGF-1 levels were similar in infants of non-diabetic mothers and mothers with gestational diabetes [285,288,289]. However, one study found sex-specific variation in cord blood IGF-1 in infants of diabetic mothers [286]: male infants had higher cord blood IGF-2, IGFBP-3, *c*-peptide and leptin and lower IGF-1 adjusted for IGFBP-3 [286], while female infants had higher cord blood IGF-1 adjusted for IGFBP-3 (Table 3) without significant differences in other hormone levels [286]. Several additional studies found elevated cord blood IGF-2 levels in infants born to mothers with gestational diabetes, but sex differences were not analyzed [118,289,290].

*12.4. Smoking*

Maternal smoking during pregnancy is associated with decreased head circumference, abdominal circumference, estimated fetal weight, and femur length (Table 2), with the latter impacted most [291,292]. The reduced femur length is detected as early as 18–24 weeks' gestation, while the smaller head circumference becomes significant around 25 weeks [291,292]. Although there is some heterogeneity among studies, reductions in abdominal circumference and estimated fetal weight emerge during the third trimester [292,293]. Disparities in head and abdominal circumference and femur length increase with gestational age [291]. When mothers stopped smoking once the pregnancy was recognized, no differences in fetal growth were seen [292]. The greatest impacts on growth were observed in the fetuses of women who had the highest tobacco use, greater than nine cigarettes per day [291,292].

Multiple studies have found decreased cord blood IGF-1 levels in infants of mothers who smoked during pregnancy (Table 3) [294–299]. Conversely, there were no associations with cord blood IGF-2, leptin, or *c*-peptide [294,295,297,298]. Studies regarding the effect of maternal smoking on cord blood adiponectin levels have yielded variable results, with some reporting no differences [295,297] and another reporting lower values [298]. Similar conflicting results were seen in cord blood levels of IGFBP-3, with some studies finding lower levels with maternal smoking [294,299] and others finding higher levels [296]. Increased cord blood ghrelin levels were found in infants of mothers who smoked [296]. Cord blood ghrelin correlated positively with the number of cigarettes smoked and negatively with birth weight [296]. Cord blood hormone levels in infants of former smokers and those exposed to passive smoke were similar to those of nonsmoking mothers [295].

*12.5. Hypertensive Disorders of Pregnancy*

Hypertensive disorders of pregnancy, which include gestational hypertension and preeclampsia, are associated with fetal growth restriction [300]; placental insufficiency is hypothesized to link the two disorders. Approximately 10% of pregnancies are affected by hypertensive disorders [301]. Compared to normotensive pregnancies, pregnancies complicated by severe preeclampsia have asymmetric fetal growth (Table 2), which manifests as decreased abdominal circumference at approximately 23 weeks of gestation [302]. Estimated fetal weight is lower by 22 weeks' gestation; reductions in fetal weight persist

through 38 weeks [302]. Compared to abdominal circumference, head circumference as well as femur and humerus lengths are less affected [302].

The effects of mild gestational hypertension and preeclampsia on fetal growth patterns are less clear. One study found a slight deceleration in estimated fetal weight and abdominal circumference in the second trimester in pregnancies complicated by mild preeclampsia, but these changes normalized later in gestation [302]. Another study found decreased head circumference and estimated fetal weight early in the third trimester in association with increased maternal diastolic blood pressure [303]. Higher diastolic blood pressure was associated with decreased head circumference, length, and weight at birth; higher maternal systolic blood pressure was also associated with decreased birth weight [303]. The effects of maternal hypertension and preeclampsia on the fetus were magnified as gestation progressed [303].

In a study that excluded women with chronic and preexisting hypertension, hypertensive disorder of pregnancy was associated with increased cord blood levels of adiponectin, leptin, and ghrelin after adjusting for gestational age, maternal age, and birth weight in preterm infants [301]. Cord blood leptin, but not adiponectin or ghrelin, was also increased in pregnancies affected by preeclampsia compared to gestational hypertension [301], Table 3. Likewise, a study comparing pregnant women without hypertension to those with mild preeclampsia and severe preeclampsia found increased cord blood leptin and decreased cord blood ghrelin in pregnancies affected by preeclampsia [304]. Conversely, a different study reported lower cord blood leptin and IGF-1 levels in pregnancies affected by preeclampsia compared to those without hypertension [305]. Finally, hypertensive disorders of pregnancy were associated with lower cord blood levels of total T4 and free T4 [287]. It is unclear if the various hormonal changes in cord blood represent adaptive responses to placental insufficiency or if they contribute directly to reductions in fetal weight gain or growth.

*12.6. Malnutrition*

Low maternal BMI early in pregnancy predisposes to fetal intrauterine growth restriction, which in turn, increases the risk of neonatal mortality [306]. Studies evaluating the impact of maternal malnutrition on fetal hormones are somewhat limited. A study conducted in Dhaka, Bangladesh compared cord blood leptin and insulin levels in infants born to women who were classified as underweight, normal weight, or overweight during the first trimester of pregnancy [307]. Rates of anemia were higher in the underweight women compared to normal weight women, but rates of vitamin D deficiency and B12 deficiency were comparable. The study found no significant differences in cord blood leptin or insulin levels among the three groups and the incidence of low birth weight was not associated with maternal BMI [307]. A study in Pune, India investigated the impact of maternal nutritional status on cord blood IGF-1 and IGFBP-3 [308]; 23% of the women had height more than two standard deviations below the mean. Average weight gain and the intake of milk and protein between 28 and 34 weeks of pregnancy fell below the Institute of Medicine's recommendations [308]. Nevertheless, cord IGF-1 and IGFBP-3 did not correlate with maternal weight, maternal BMI at 28 or 34 weeks, maternal weight gain from 28 to 34 weeks, or maternal protein or fat intake at 28 or 34 weeks [308]. These observations demonstrate the power of maternal and placental adaptations to provide nutritional support to the fetus even in the face of limited maternal bodily energy reserves. On the other hand, the study found a positive association between maternal milk intake at 34 weeks and cord blood IGF-1 levels [308]. Cord IGF-1 correlated positively with birth weight, ponderal index, and abdominal circumference, while IGFBP-3 was positively correlated with ponderal index. Neither cord IGF-1 nor IGFBP-3 levels correlated with birth length [308].

Nutritional anemia is common in low- and middle-income countries and serves as a marker of malnutrition. A study [309] based in New Delhi, India found that of the 75% of pregnant women with anemia, 85% had moderate anemia (Hb > 7–10.9 g/dL) and 15%

had severe anemia (Hb < 7 g/dL). Pregnant women in the severe anemia group had low pre-pregnancy weight and their infants had lower birth weights. Cord blood IGF-1 levels were increased in infants of anemic mothers (Table 3) compared to infants of mothers with normal blood counts [309]. Additionally, cord blood T4 levels were decreased in the moderate and severe anemia groups compared to the non-anemic group. The severe anemia group also had significantly higher levels of cord blood prolactin, insulin, and placental lactogen compared to the moderate anemia group, but these were not significantly different from the non-anemic group [309].

## 13. Effects of Maternal Disorders on Placental Nutrient Delivery

The insulin resistance of pregnancy is exacerbated by maternal obesity, excessive gestational weight gain, and gestational diabetes, resulting in even lower maternal levels of adiponectin and higher levels of insulin, leptin, and IGF-1 [310]. Additionally, affected mothers generally have elevated serum levels of glucose, triglycerides, and free fatty acids [310]. Notably, the placenta maintains insulin sensitivity even in the setting of increasing maternal insulin resistance; thus, the elevated insulin levels found in maternal obesity, excessive gestational weight gain, and gestational diabetes drive placental insulin and mTOR signaling, leading to increased glucose and amino acid transport to the fetus [310]. High maternal leptin and low adiponectin similarly promote placental amino acid transport, while elevated levels of IGF-1 increase both glucose transport and amino acid uptake in the placenta [310]. Together, these factors enhance nutrient transport to the fetus and thereby promote fetal overgrowth.

The effects of maternal hormones on placental nutrient transport in hypertensive disorders of pregnancy and maternal smoking are less well understood. Maternal smoking directly impacts placental health by decreasing placental blood flow and oxygenation [311]. Expression of placental amino acid and fatty acid transporters are variably decreased and glucose transporters variably increased in smoking-exposed pregnancies [312]. Maternal smoking does not appear to alter maternal leptin levels, although placental GH levels are decreased [313]. Additionally, pregnant women who smoke have lower IGF-1/IGFBP3 ratios compared to those who do not smoke [313]. Placental dysfunction in pre-eclampsia causes systemic inflammation and maternal endothelial dysfunction [314]. In preterm preeclampsia (<37 weeks' gestation), placental hypoperfusion leads to placental hypoxia and ischemia and limits fetal nutrient availability [314].

## 14. Effects of Preterm Delivery on Neonatal Hormones, Growth Factors, and Adipocytokines

Delivery of an infant prior to term disrupts the maturation of pathways controlling the production of hormones, growth factors, and adipocytokines and may trigger hormonal adaptive responses to stress and illness that have profound implications for extrauterine growth, weight gain, and metabolic function. For example, immaturity of pancreatic beta cell development can limit neonatal insulin production, while the stress of preterm delivery, neonatal illness, and pharmacologic interventions (e.g., diuretics and dopamine) stimulate increases in glucocorticoids and catecholamines and may suppress the activity of the hypothalamic–pituitary–thyroid axis.

Disruptions in hormone homeostasis are compounded by nutritional deficits owing to the loss of the placenta and (in some cases) immaturity and inflammation of the gastrointestinal tract. As shown in Figure 12, the levels of **IGF-1** are considerably higher in normal infants in utero than in infants born preterm and matched for post-conceptual age [91]. Thus, infants born prematurely have a relative IGF-1 deficiency.

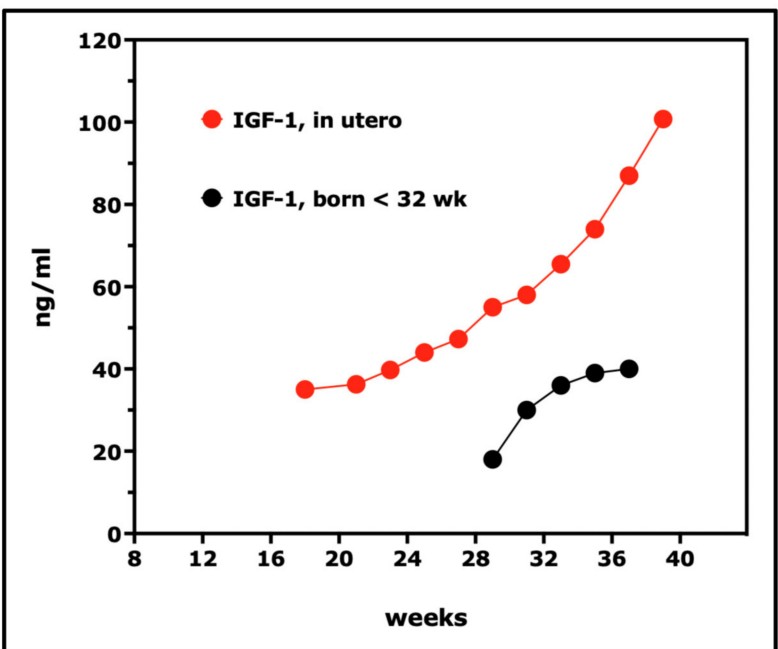

**Figure 12.** Serum IGF-1 in the human fetus in utero and in infants born prematurely and assayed at the equivalent post-conceptual age. The figure is adapted from data presented in Hellstrom A et al. [91].

Likewise, levels of **leptin** are markedly higher in normal infants in utero than in infants born preterm and matched for post-conceptual age (Figure 13). Thus, infants born prematurely have a relative (and severe) deficiency of leptin as well as IGF-1 [315]; as with IGF-1, the magnitude of the deficit in leptin in the preterm infant is inversely proportional to post-conceptual age. At 36 weeks post-conceptual age, leptin concentrations are higher in girls than boys [315], likely reflecting sex differences in extrauterine fat deposition.

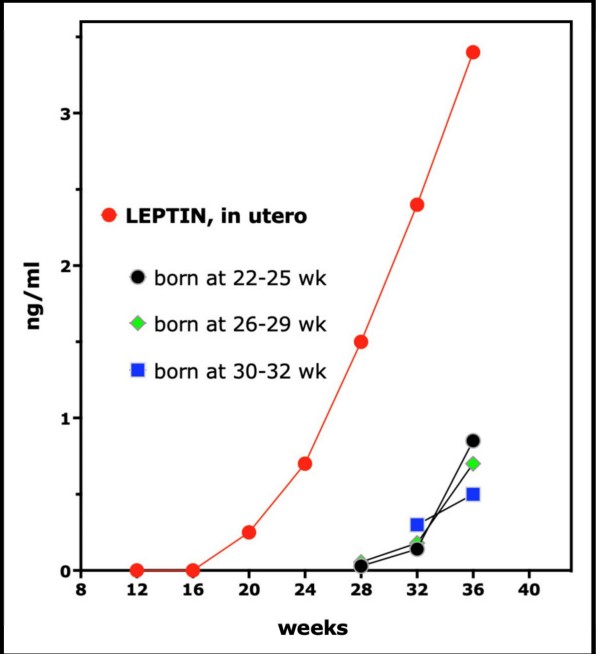

**Figure 13.** Serum leptin in the human fetus in utero and in infants born prematurely and assayed at the equivalent post-conceptual age. The figure is adapted from data presented in Steinbrekera B et al. [315].

Infants born prematurely also have low levels of free **T4**, **T3**, and **TSH** compared with infants born at term (Figure 14). The TSH surge at birth is blunted in preterm infants and free T4 and T3 levels remain low for more than a month after delivery [185,316].

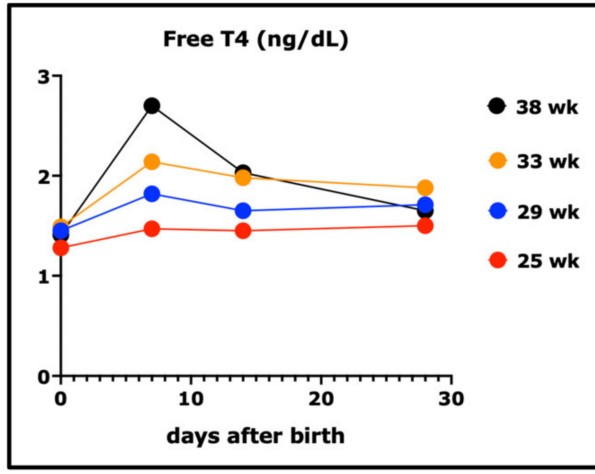 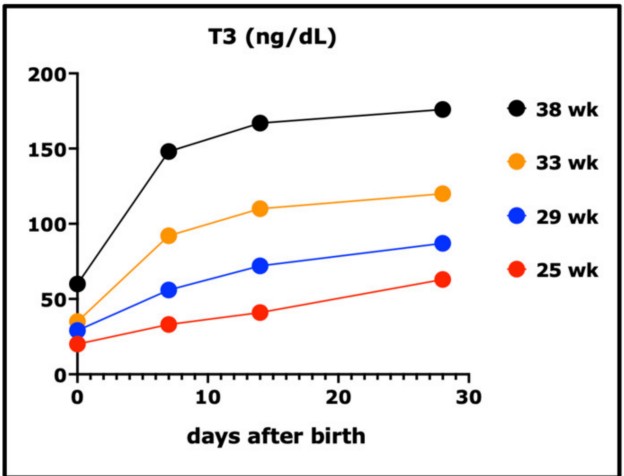

**Figure 14.** Thyroid hormones in preterm and term (38 weeks) infants after birth. The figure is adapted from data in Williams FL et al. [316].

## 15. Implications of Preterm Birth for Neonatal Growth, Weight Gain, and Metabolic Function

Disruptions in hormone and growth factor production and hormonal adaptive responses to stress, illness, and malnutrition have profound effects on growth, weight gain, and metabolic function during the neonatal and early postnatal periods (Figure 15).

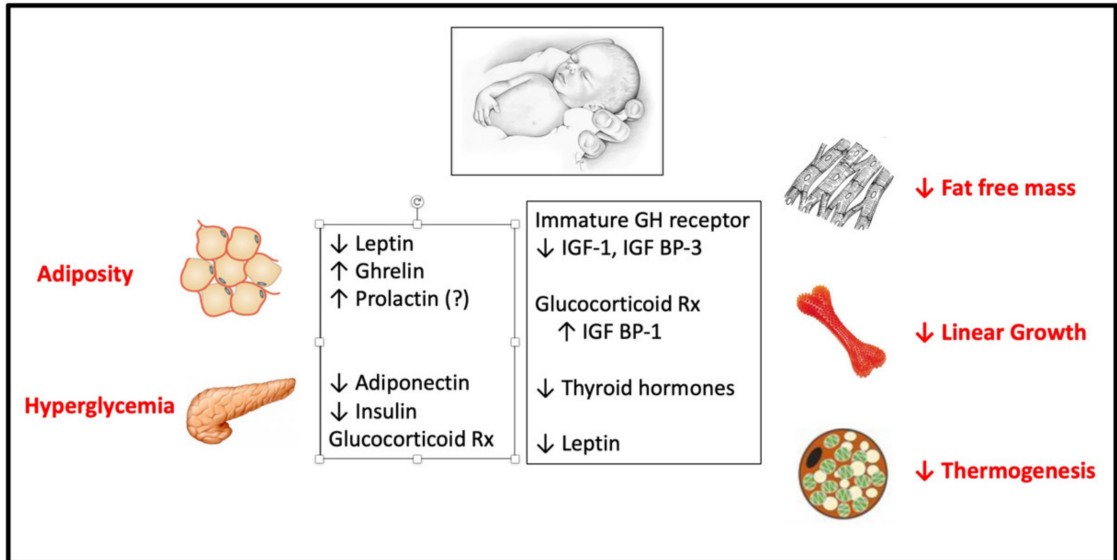

**Figure 15.** Effects of preterm birth on linear growth, fat deposition, energy expenditure and metabolic balance.

The patterns and relative rates of postnatal weight gain, fat deposition, and linear growth in infants born preterm differ from the patterns and relative rates of weight gain, fat deposition, and linear growth in utero in healthy infants born at term. This was best demonstrated in a Swedish study by Chmielewska et al. [317], but similar results have been found in other investigations [43,318]. Relative to infants born at term, infants born at a mean of 28 weeks' gestation had sharp reductions in weight and length z-scores during the first 2–5 weeks of age. The initial reduction in length z (nadir −2.1 to 3.5 SD) was more

striking than the reduction in weight z (nadir −2.2 SD). At term equivalent age, weight z in the preterm infants was −0.9 SD, while length z was −1.8 SD. Thus, the rate of catch-up weight gain in the preterm infants exceeded the rate of gain in linear growth: preterm infants at term equivalent age have higher total fat mass and percent body fat (preterm 20.2%, term AGA 11.7%) and lower fat-free/lean body mass and % fat-free mass (Figure 16). It is possible that early accrual of fat in the preterm infant represents an adaptive mechanism designed to promote survival under stress [108]. Reductions in length z and fat-free mass in the preterm infant persist through 4 months of postnatal age, while fat mass and % body fat are normalized [43,317,318].

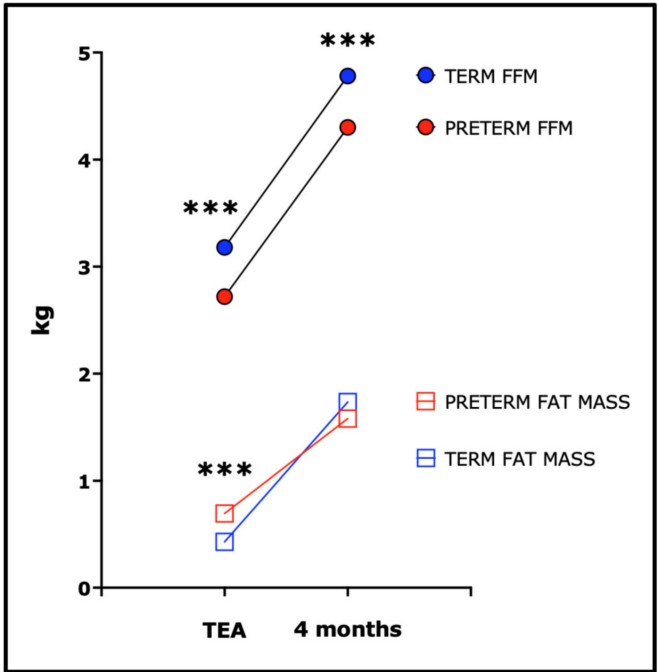

**Figure 16.** Fat mass and fat-free mass (FFM) in term and preterm infants at birth (term infants) and at term equivalent age (TEA, preterm infants) and 4 months later. The figure is adapted from data in Chmielewska et al. [317]; qualitatively similar results were found in other investigations (Hamatschek C et al. [43] and Casirati A et al. [318]). *** significantly different, $p < 0.001$.

The relative deficits in linear growth and fat-free mass in preterm infants likely reflect the immaturity of the hepatic GH receptor and the relative deficiencies of IGF-1, IGFBP-3, and thyroid hormones: the rate of long bone growth in preterm infants after birth correlates positively with changes in IGF-1 (and to a lesser extent IGFBP-3) concentrations [319]. High levels of IGFBP-1 may contribute to growth failure in critically ill or malnourished infants and/or those treated with high doses of glucocorticoids [92,320,321]. Extreme variability (and often undetectability) of FGF21 levels in the neonatal period makes its relevance for postnatal (as well as fetal) growth unclear [112,322].

Immaturity of beta cell function in the very low birth weight preterm infant can limit insulin production and impair glucose tolerance in the neonatal period. Neonatal hyperglycemia is in some cases potentiated by insulin resistance, a consequence of illness, infection, glucocorticoid excess, and a relative deficiency of adiponectin.

Low concentrations of leptin [315] and thyroid hormones [316] in the preterm infant at birth reduce sympathetic nervous system activity and impede brown adipose thermogenesis. Together with a lack of insulating white adipose stores, these predispose to neonatal hypothermia. The low levels of leptin and high levels of prolactin [136,323], together with a rise acylated ghrelin [324,325] may contribute to weight re-gain and may explain in part the subsequent (relative) adiposity of preterm infants: hypoleptinemia is associated with higher rates of weight gain in infancy [326–328]. Leptin concentrations in preterm as well

as term infants initially fall after delivery owing to the loss of the placenta and the transient lack of nutrient intake. However, leptin levels in infants born preterm rise thereafter in association with increases in body weight and correlate positively with the ratio between weight and length [329,330].

As noted previously, preterm infants have a relative deficiency of adiponectin at birth [221,239]. Adiponectin levels in full term infants decline sharply from birth through age 2 years [331]. Conversely, adiponectin levels in low-birth-weight preterm infants rise steeply during the first few weeks of life, peaking at 40 weeks post-conceptual age [332]. Limited evidence [331] suggests that low adiponectin levels in early childhood are associated with increases in plasma triglycerides and low levels of HDL, features suggesting insulin resistance. Yet cord blood triglyceride levels correlated negatively with weight gain and BMIz at 6–18 months in a prospective study of 2267 Chinese infants [333], and postnatal weight gain in preterm infants is associated more strongly with increases in leptin than with reductions in adiponectin [334].

## 16. Effects of Nutritional and Pharmacologic Interventions on Growth and Weight Gain in Preterm Infants: Correlation with Hormones, Growth Factors, and Adipocytokines

### 16.1. Breast Feeding and Breast Milk Hormones

Providing adequate nutrition is critical to support healthy growth and development. Human milk is the optimal source of enteral nutrition for preterm infants, and pasteurized donor human milk is recommended for very low birth weight infants when mother's own milk is not available [335–337]. Human milk confers numerous benefits in preterm infants including protection against necrotizing enterocolitis and sepsis. However, human milk requires fortification to provide sufficient protein, energy, and minerals to meet nutritional requirements. Total energy intake and total protein intake correlate with serum IGF-1 in preterm infants [338–340]. Total energy intake also correlates with serum IGFBP-3, while protein intake correlates inversely with IGFBP-2 [339]. Interestingly, preterm infants fed preterm formula have higher in-hospital rates of weight gain and head growth than preterm infants fed human milk [341]. And in term infants, breastfeeding is associated with lower infant IGF-1 levels compared to formula feeding [342]. The effects of diet and specific nutrients on other hormonal mediators of postnatal growth in preterm infants have not been studied extensively. One investigation reported that leptin levels correlate with human milk intake in preterm infants [343].

Human milk contains an abundance of hormones, cytokines, and growth factors including leptin, adiponectin, ghrelin, prolactin, calcitonin, erythropoietin, epidermal growth factor, FGF21, IGF-1, and IGF-2 [344,345]. Numerous studies have reported associations between hormone levels in milk and infant hormone levels and growth. For example, leptin levels in preterm infants correlate with breast milk leptin levels [346]. One study reported that milk leptin concentrations on postnatal day 5 were positively associated with fat mass at NICU discharge, but negatively associated with fat mass at 4 months' corrected age [347]. Another study found that total milk leptin intake in the first 28 postnatal days was positively associated with weight gain velocity and weight z-score at birth to 36 weeks' postmenstrual age (PMA) [348]. In this study, adiponectin intake from milk was associated with greater length z-scores but lower head circumference z-scores at 36 weeks' PMA. Another study found no correlation between milk adiponectin and growth outcomes [347]. Milk IGF-1 concentrations are associated with fat-free mass at 4 months' corrected age, while milk ghrelin concentrations are associated with neonatal weight gain and linear growth [347,349].

The effects of breast milk hormones on neonatal growth and weight gain could in theory be mediated through systemic absorption of the intact peptides across an immature gastrointestinal barrier. In fact, undegraded milk peptides localize predominantly to the wall of the stomach and small intestine. However, a few studies show limited transfer of enteral leptin, IGF-1, and prolactin [350–354] in newborn piglets and rats; systemic uptake wanes quickly over the first few weeks of life.

Alternatively, breast milk hormones might regulate infant weight gain and growth through direct effects on the growth, integrity, and/or function of the neonatal gastrointestinal tract [355–357]. For example, one study showed that leptin administered in formula for 6 days to neonatal piglets increased small-intestinal length and the activities of intestinal enzymes [357]. Oral administration of leptin to newborn rats, however, had no effect on body weight or fat stores [351]. Likewise, enteral administration of pharmacologic doses of IGF-1 to neonatal piglets and suckling mice increased small intestinal mucosal growth [353]. On the other hand, supplementation of formula with IGF-1 at concentrations twice those of human colostrum had no effect on weight gain, knee-heel length, calf circumference, skin-fold thickness, or gastrointestinal carbohydrate absorption in preterm infants with birth weights 750–1250 g [358]. Thus, the roles of breast milk hormones, cytokines, and growth factors in human neonatal growth and weight gain remain unclear. Breast milk hormone concentrations may represent biomarkers of maternal nutrition, BMI, or other factors that influence infant growth, while concentrations of leptin and IGF-1 in the infant may simply reflect the adequacy of breast feeding and nutrient intake and the rate of postnatal fat deposition.

*16.2. Glucocorticoid Therapy*

Hydrocortisone and dexamethasone are among the 20 most commonly prescribed medications for extremely low birth weight preterm infants in the NICU [359]. Indications for use of glucocorticoids in preterm infants include refractory hypotension and prevention and treatment of chronic lung disease. Despite beneficial effects on pulmonary outcomes in high-risk preterm infants, dexamethasone use is associated with adverse short- and long-term complications including an increased risk of cerebral palsy [360,361]. Short-term adverse effects of dexamethasone treatment include hyperglycemia, muscle catabolism, and poor in-hospital growth [362–364]. Dexamethasone causes transient increases in leptin and insulin, marked suppression of the hypothalamic–pituitary–adrenal axis, and decreased activity of the GH-IGF axis, with concomitant inhibitory effects on linear growth [365–367]. A randomized trial of dexamethasone for chronic lung disease of prematurity found that at 2-year follow-up, boys treated with dexamethasone had lower weight and height than untreated controls [368]. At school-age follow-up, children treated with dexamethasone had lower height and smaller head circumference than controls [361]. In an observational study, postnatal dexamethasone exposure was negatively associated with length, weight, weight-for-length, and head circumference at 2 years corrected age [369]. Another study found that neonatal dexamethasone treatment was associated with lower height, weight, and bone mass at prepubertal age [370].

In contrast to dexamethasone, hydrocortisone use at doses ranging from 1 to 5 mg/kg/day has limited—and in some cases, no—effect on somatic growth [371–376]. For example, a randomized trial of early low-dose hydrocortisone showed no effect on weight gain or head growth at 36 weeks' postmenstrual age (PMA) [376]. Likewise, preterm infants treated for 22 days with hydrocortisone had higher weight at 36 weeks PMA than placebo-treated controls; length and head circumference were not significantly different between groups [375]. A randomized trial of a 10-day course of hydrocortisone vs. placebo found no significant differences in weight, length, or head circumference z-scores at 36 weeks' PMA or 22–26 months' corrected age [372]. In contrast, an observational study of preterm infants found that both hydrocortisone and dexamethasone therapy in the NICU were associated with significant reductions in weight and height standard deviation scores and, to a lesser extent, head circumference at 2–5 months of age [377]. The initial decrease in growth velocity was followed by a rebound increase in growth rate after treatment and a delay in the age at which maximal growth velocity of weight/size was reached. These effects were more pronounced in infants treated with dexamethasone than with hydrocortisone. Interpretation of these findings, like numerous others in the field, is challenging because "untreated controls" were born at later gestational age, at higher birth weight, and with fewer and less serious co-morbidities including ventilator-dependent pulmonary disease.

It should also be noted that short-term treatment with hydrocortisone at doses exceeding 5 mg/kg/d or prolonged courses of moderate- or high-dose hydrocortisone could have adverse effects in the preterm infant similar to those of the more potent dexamethasone.

## 17. Summary

Linear growth, weight gain, and fat storage in the fetal and early postnatal periods are controlled by macronutrient availability and a complex interplay of hormones, cytokines, and growth factors produced by the pregnant mother, placenta, fetus, and newborn infant. Pathologic conditions impacting the pregnant mother can alter the delivery and utilization of macronutrients and the production and action of placental, fetal, and neonatal hormones and can thereby modify the course of fetal and postnatal growth. Infants born preterm have deficiencies of critical hormones and growth factors that normally emerge or increase in late gestation and undergo adaptive hormonal responses that influence postnatal growth and the accrual of lean and fat mass. Growth and weight gain in the preterm infant are subject to mode of feeding and to potential deleterious effects of high-dose glucocorticoids.

## 18. Gaps in Knowledge

Despite the wealth of information accrued during the past 50 years, there remain major gaps in our understanding of the control of human fetal and early postnatal growth and the mechanisms contributing to long-term metabolic risk in preterm infants. Important questions include:

1. How do sex differences in fetal and postnatal growth and weight gain influence the risks of future metabolic and cardiovascular disease in infants born preterm?
2. Do the lactogenic hormones or growth hormone increase beta cell mass or insulin production in the human fetus? Might other hormones or growth factors serve as fetal beta cell tropins?
3. Do macronutrients regulate human fetal IGF production during early and mid-gestation? How is this control exerted? What are the roles of lactogenic hormones in the control of IGF production in the human fetus?
4. Do fetal adipocytokines or ghrelin directly modulate human fetal growth, weight gain, or metabolic function?
5. What are the effects of maternal pathologies on human fetal hormones, cytokines, and growth factors during the course of intrauterine development? How would this information help us to understand the pathogenesis of growth failure or excess adiposity in affected preterm infants and their potential risks for long-term complications?
6. Do hormones, cytokines, and growth factors in breast milk modulate the growth or weight gain of the preterm or term infant? How are these effects mediated?
7. How should we optimize nutrition to promote growth and cognitive development in preterm infants while limiting long term cardiovascular and metabolic risks? Does the preferential storage of fat relative to skeletal muscle and bone growth provide a survival advantage for the preterm infant? Might there be an association between neonatal fat storage, energy availability, and long-term cognitive function?

**Author Contributions:** L.P. reviewed the published literature and contributed to writing the manuscript. N.Y. reviewed the published literature and contributed to writing the manuscript. M.F. was invited to submit the manuscript; he reviewed the published literature and contributed to writing the paper. All authors have read and agreed to the published version of the manuscript.

**Funding:** This research received no external funding. Noelle Younge receives support from the National Institutes of Health (K23DK120960, R03DK134684) and the Zeist Foundation. Michael Freemark receives support from Rhythm Inc for a study of monogenic obesity.

**Institutional Review Board Statement:** The study did not require ethical approval.

**Informed Consent Statement:** Not applicable.

**Data Availability Statement:** No new data generated.

**Acknowledgments:** The authors thank the Duke Divisions of Pediatric Endocrinology and Neonatology and the Duke Molecular Physiology Institute for academic support.

**Conflicts of Interest:** The authors declare no conflict of interest.

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
