# Peer review of "Hormonal Determinants of Growth and Weight Gain in the Human Fetus and Preterm Infant"

_nutrients, doi:10.3390/nu15184041_

Round 1
Reviewer 1 Report
The topic is relevant and the manuscript is well written. Tables and figures are appropriate and facilitate understanding of the subject.
However, the text requires adjustments: what was the methodology used to select the articles? What was the search period, database, keywords, criteria for inclusion or exclusion? The text is long and has many pages and topics. I suggest inserting a summary or an index to make it easier to find the different subjects covered.
Author Response
Reviewer 1:
The topic is relevant and the manuscript is well written. Tables and figures are appropriate and facilitate understanding of the subject.
However, the text requires adjustments: what was the methodology used to select the articles? What was the search period, database, keywords, criteria for inclusion or exclusion? The text is long and has many pages and topics. I suggest inserting a summary or an index to make it easier to find the different subjects covered.
Response:
Thank you for your positive comments and review of the paper.
We have expanded the section on Methodology to include information requested; see revised page 3, lines 93-136
We have included an Index (revised page 1) as well as a Summary (revised page 35)
Other changes are highlighted in yellow

Reviewer 2 Report
This is a well-written and scholarly treatise which summarizes a huge body of work. I have only a few minor suggestions.
1. Could the authors make clear the difference between adipogeneis and lipogenesis?
2. In the legend of Figures 12 and 13 there is mention of "equivalent post-menarcheal age". I am pretty sure they mean "post-conceptual (or perhaps post-menstrual) age"
3. On Page 31 there is a discussion of the effects of hormones and growth factors in human milk. Could the authors comment on whether there is evidence that protein hormones and growth factors can be absorbed intact from the infant intestines?
4. There is so much in this article that it took me a long time to get through it. I felt that at the end, either before or after the "gaps in knowledge section", that there be a "Conclusions"or "key takeaways" section in which the authors concisely summarize what they feel to be the 8-12 most important lessons which have been learned over the past few decades on this fascinating subject.
Author Response
Reviewer 2:
This is a well-written and scholarly treatise which summarizes a huge body of work. I have only a few minor suggestions.
- Could the authors make clear the difference between adipogeneis and lipogenesis?
- In the legend of Figures 12 and 13 there is mention of "equivalent post-menarcheal age". I am pretty sure they mean "post-conceptual (or perhaps post-menstrual) age"
- On Page 31 there is a discussion of the effects of hormones and growth factors in human milk. Could the authors comment on whether there is evidence that protein hormones and growth factors can be absorbed intact from the infant intestines?
- There is so much in this article that it took me a long time to get through it. I felt that at the end, either before or after the "gaps in knowledge section", that there be a "Conclusions"or "key takeaways" section in which the authors concisely summarize what they feel to be the 8-12 most important lessons which have been learned over the past few decades on this fascinating subject.
Response:
Thank you for your positive comments and review of the paper.
We clarify the difference between adipogenesis and lipogenesis in the revised narrative page 5, lines 178-180
Thank you for your comment on the legends to Figures 12 and 13; this has been changed to “post-conceptual age”
We now include an expanded section on the effects and systemic uptake of breast milk hormones; see the revised manuscript pages 33-34, lines 737-775
We have included an Index (revised page 1) as well as a Summary (revised page 35); given the breadth of the discussion, the length of the manuscript and our current list of Gaps in Knowledge, we think the Summary is preferable to a list of “lessons”
Other revisions are highlighted in yellow
